# New insight into 3D mesoscale eddy properties from CMEMS operational models in the western Mediterranean

Evan Mason[1,2], Simón Ruiz[1], Romain Bourdalle-Badie[3], Guillaume Reffray[3], Marcos García-Sotillo[4], and Ananda Pascual[1]

[1]IMEDEA, Esporles, Mallorca, 07190, Spain
[2]Applied Physics Laboratory, University of Washington, Seattle, WA 98105, USA
[3]Mercator Ocean, 8-10 Rue Hermès, 31520, Ramonville Saint-Agne, France
[4]Puertos del Estado, Madrid, 28042, Spain

*Correspondence to:* Evan Mason (emason@imedea.uib-csic.es)

**Abstract.** Rapid evolution of operational ocean forecasting systems is driven by advances in numerics and data assimilation schemes, and increase of *in situ* and satellite observations. The Copernicus Marine Service (CMEMS) is a major provider of operational products that are made available through an online catalogue. The service includes global and regional forecasts in near-real-time and reanalysis modes. Here we apply an eddy tracker to daily SSH fields from three such reanalysis products from the CMEMS catalogue, with the objective to evaluate their performance in terms of their eddy properties and three-dimensional composite structures over the period 2013 through 2016. The products are (i) the Global Analysis Forecast, (ii) the Mediterranean Analysis Forecast, and (iii) the Iberia-Biscay-Ireland Analysis Forecast. The common domain between these reanalyses is the western Mediterranean Sea (WMED) between the Strait of Gibraltar and Sardinia. This is a complex region with strong density gradients, especially in the Alboran Sea in the west where Atlantic and Mediterranean waters compete. Surface eddy property maps over the WMED of eddy radii, amplitudes and nonlinearity are consistent between the models, as well as with gridded altimetric data that serve as a reference. Mean 3D eddy composites are shown only for three subregions in the Alboran Sea. These are mostly consistent between the models, with minor differences being attributed to details of the respective model configurations. This information can be informative for the ongoing development of these CMEMS operational modeling systems. The mesoscale data provided here may be of interest to CMEMS users and could in the future be a useful addition to a more diverse CMEMS catalogue.

## 1   Introduction

The Copernicus Marine Environment Monitoring Service (CMEMS) supplies information about the physical state and variability of the global ocean and regional seas (von Schukmann et al., 2016; Le Traon et al., 2017).
CMEMS distributes remote sensing and *in situ* observations, and short-term model forecasts in response to the needs of European public and private users. The CMEMS architecture includes seven Monitoring and Forecasting Centers (MFC) that generate operational products such as short-term forecasts, hindcasts and reanalyses in different areas of the European seas (Arctic Ocean, Baltic Sea, northwest European shelves, Iberia-Biscay-Ireland area, and the Mediterranean and Black Seas). Additionally, there is a global MFC that
delivers products at global ocean scale. A detailed description of each of the seven MFCs can be found in Le Traon et al. (2017). The quality of the products from these operational systems is crucial because they are used, together with observations, to detect and analyse environmental variability and trends (von Schuckmann et al., 2018). In this sense, a continuous effort to improve CMEMS products is made through new research and development projects funded by the CMEMS service evolution (Le Traon et al., 2019).

Three CMEMS MFCs produce short-term forecasts for the entire or partial Mediterranean Sea: (i) the Global Mercator model (*GLO*), (ii) the Mediterranean Forecasting System (*MFS*), and (iii) the Iberia-Biscay-Ireland system (*IBI*); the latter covers only the western Mediterranean (WMED). As in other regions of the European seas, improvement of the short-term forecasts in the WMED has been a priority for the Service Evolution. To develop the service and produce better forecasts of the ocean in this particular area and in the
global ocean in general, we need to increase our understanding of 2D and 3D ocean circulation, dynamics and interactions at different scales, namely the mesoscale (10-100 km) and fine scale (1-10 km). Simulating dynamics at these scales with numerical models is challenging; trade-offs are made between the need for accurate representation of topography and grid resolution (both of which impact volume transport); impact of inclusion of tidal forcing; the need for assimilation, among other factors.

Our objective here is to evaluate the performance of three CMEMS operational oceanic models in the WMED using a subregional three-dimensional (3D) eddy-centric compositing approach. We use an eddy tracker to identify daily positions and sizes (radius, amplitude) of mesoscale eddies in each model solution

(Sec. 3). This information allows estimation of indices to extract 2- and 3D arrays of data (e.g., sea surface height, temperature, etc.) that extend a horizontal distance well beyond the eddy radius, and from the surface to the ocean floor. Selective averaging of these data *cubes* allows generation of mean eddy signals for these variables over predefined subregions (e.g., Mason et al., 2017). Previous work with eddy composites built from hundreds to thousands of eddy observations has contributed to better understanding of the relationships between eddies and, for example, sea surface chlorophyll (e.g., Chelton et al., 2011a; Gaube et al., 2014), sea surface temperature (SST; e.g., Hausmann and Czaja, 2012), surface heat fluxes (e.g., Villas-Bôas et al., 2015), and ocean winds (e.g., Frenger et al., 2013). For a comprehensive review of this topic see McGillicuddy (2016).

The paper is structured as follows. We provide a brief review of the western Mediterranean study region in Sec. 2, focusing in particular on the Alboran Sea. The eddy tracking and compositing methodology is described in Sec. 3. Results in Sec. 4 comprise an analysis of eddy properties from the models and altimetry over the western Mediterranean, followed by a subregional 3D eddy compositing analysis that is focused on the Alboran Sea. (Results from other subregions in the WMED are included in Secs. A3 and A4). A discussion of the results and final concluding remarks are made in Sec. 5.

## 2 The western Mediterranean study region

The Mediterranean Sea is often described as an easily-accessible reduced-scale ocean laboratory which hosts almost all of the physical phenomena found in different regions of the global ocean. These processes, which include deep convection (Leaman and Schott, 1991; Herrmann et al., 2009), shelf-slope exchange (Bethoux and Gentili, 1999), thermohaline circulation and water mass interaction (Bergamasco and Malanotte-Rizzoli, 2011, and references therein), mesoscale (Robinson and Golnaraghi, 1994) and submesoscale dynamics (Bosse et al., 2015), can be sampled and investigated at smaller scales.

In the western Mediterranean (Fig. 1), the Alboran Sea is characterised by the presence of two anticyclonic gyres (Vargas-Yáñez et al., 2002; Renault et al., 2012), and their associated strong fronts that are mostly governed by salinity (Tintoré et al., 1988; Allen et al., 2001). This sea is the most energetic region of the western Mediterranean (e.g., Pascual et al., 2007; Capó et al., 2019). The topography in the Alboran Sea is steep to the north, west and south (see inset in Fig. 1) with maximum gradients of above 25° (e.g., Costello

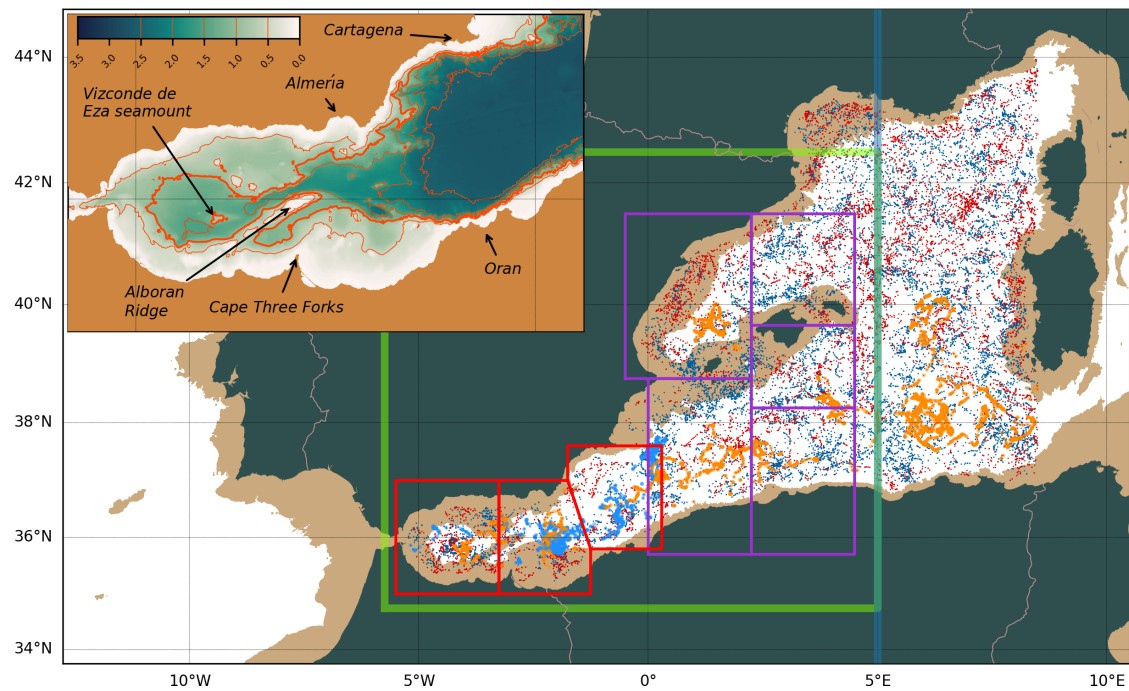

**Figure 1.** Map of the western Mediterranean study region. Cyclonic and anticyclonic eddy tracks from gridded altimetry are plotted in blue and red, respectively, for the period 2013 to 2016; tracks with lifetimes greater than one year are highlighted in light blue and orange. Brown shading indicates bathymetry shallower than 1000 m. The green box shows the domain used for the eddy property analysis in Sec. 4.1. The blue line marks the eastern boundary of the *IBI* model domain. The box areas bounded in red in the Alboran Sea indicate the three subregional domains used for eddy compositing in Sec. 4.2. Bounded areas in purple show five additional subregions for which composites are shown in Secs. A3 and A4. The inset map provides further detail of the topography and relevant physical features of the Alboran Sea and coast.

et al., 2010). A shallow undersea ridge known as the Alboran Ridge extends northeastward from Cape Three Forks towards the center of the EAG; the 0.7 km$^2$ Alboran Island is found on the ridge at -3.0°W, 35.9°N. The Alboran Trough is a deep water channel along the northern base of the ridge that connects the western and eastern basins. Mass exchange with the north Atlantic takes place at the open Strait of Gibraltar to the west, and to the east with the wider western Mediterranean. In the transition region between the Alboran Sea and the Algerian sub-basin, intense eddies and fronts are also generated although they are less frequent than in the Alboran Sea (Pascual et al., 2017). The presence of large eddies in the Algerian basin has been systematically documented (e.g., Ruiz et al., 2001; Puillat et al., 2002; Escudier et al., 2016; Pessini et al., 2018). These Algerian eddies typically form as a result of instabilities in the cool and fresh Algerian coastal current (e.g., Millot, 1999; Testor et al., 2005; Capó et al., 2019). Recent studies in this basin using high-resolution observations (Cotroneo et al., 2016; Aulicino et al., 2018) have demonstrated the presence of fine-scale features associated with the large eddies. Regarding the Balearic Sea, the spatial-temporal variability of the surface circulation was investigated by Mason and Pascual (2013) revealing intense mesoscale eddy activity in this northern sub-basin.

## 3 Data and methods

### 3.1 The CMEMS models

The stated general objectives for the CMEMS MFCs is to produce near-real-time short-term (5-10 days) forecasts of currents and other oceanographic variables such as temperature, salinity and sea level, that will enable quicker responses to oil spills and other emergencies at sea, as well as support for efforts to achieve better understanding of ocean dynamics. With each forecast product, namely *GLO*, *MFS* and *IBI*, there is also an associated historical reanalysis, and these are the solutions that we work with here.

The numerical code used for each of these CMEMS models is the Nucleus for European Modeling of the Ocean (NEMO, e.g., Madec, 2008). NEMO solves the three-dimensional finite-difference primitive equations in spherical coordinates on an Arakawa-C grid and a vertical $z$-coordinate scheme. It assumes hydrostatic equilibrium and the Boussinesq approximation and makes use of a non-linear split-explicit free surface to simulate fast external gravity waves such as tidal motions. Steep slopes (common in the semi-enclosed

**Table 1.** Main characteristics of the three CMEMS models. Model resolutions are given in degrees; equivalents in km for the Mediterranean are included in parentheses).

|  | *GLO* | *MFS* | *IBI* |
|---|---|---|---|
| Resolution (°) | $1/12$ (∼7 km) | $1/16$ (∼4 km) | $1/36$ (∼2 km) |
| Vertical levels | 50 | 72 | 50 |
| Surface forcing | 3-hourly; ECMWF | 6-hourly; ECMWF | 3-hourly; ECMWF |
| Boundary forcing | N/A | *GLO* | *GLO* |
| Rivers | Dai et al. (2009) database | Global Runoff Data Centre (Fekete et al., 1999) | Daily/monthly blend (Maraldi et al., 2013) |
| Version NEMO | 3.1 | 3.6 | 3.6 |
| Tides | No | No | Yes |
| Data assimilation | Yes (T/S, SLA, SST) | Yes (T/S, SLA, SST) | No |
| DA-MDT | Yes | Yes | N/A |
| Ice | Yes | N/A | N/A |
| Topography | GEBCO08 < 200 m, ETOPO1 > 300 m | GEBCO30 | GEBCO08 |

western Mediterranean) are well resolved by the use of partial bottom cells to represent the bathymetry (e.g., Barnier et al., 2006).

The main characteristics of these products and their differences are listed in Tab. 1. Daily mean prognostic variables for the period 1 January 2013 to 30 June 2016 from each product were downloaded by *ftp* from the Copernicus CMEMS portal (http://marine.copernicus.eu/). Although the models are run on irregular grids the MFCs, for the convenience of users, provide the data on regular grids. Further details about each model are provided in the following subsections.

### 3.1.1 Mercator Global (*GLO*; GLOBAL_ANALYSIS_FORECAST_PHY_001_024)

The Mercator Global Ocean forecasting system is produced by Mercator Ocean (France) (e.g., Hernandez et al., 2015; Gasparin et al., 2018; Lellouche et al., 2018). The simulation has a horizontal resolution of 1/12°

and 50 vertical $z$-levels. The model includes multivariate data assimilation, which consists of a Singular Extended Evolutive Kalman (SEEK) filter analysis of along-track satellite SLA and SST together with *in situ* profiles of temperature and salinity. The altimeter reference period for the assimilated SLA is 20 years (Rio et al., 2014). The assimilated SST is taken from the CMEMS TAC daily level-4 OSTIA composite product (Donlon et al., 2012). In the Gibraltar Strait there is relaxation of temperature and salinity towards Levitus 2013 values (Locarnini et al., 2013; Zweng et al., 2013). The bathymetry used in the system is a combination of the ETOPO1 (Amante and Eakins, 2009) and GEBCO08 (Becker et al., 2009) topography databases: ETOPO1 (GEBCO08) is used in regions deeper (shallower) than 300 (200) m with linear interpolation over the 200-300 m layer.

### 3.1.2    Mediterranean Forecast System (*MFS*; MEDSEA_ANALYSIS_FORECAST_PHYS_006_001)

*MFS* is a product of the Italian Mediterranean Forecasting System. Its horizontal resolution is 1/16°, with 72 unevenly spaced vertical $z$-levels. *MFS* includes data assimilation (based on an OceanVAR scheme) of temperature and salinity vertical profiles, satellite SST and along-track satellite SLA observations (Dobricic and Pinardi, 2008; Dombrowsky et al., 2009; Tonani et al., 2015). The 20-year mean dynamic topography of Rio et al. (2014) is used for the assimilation of along-track SLA.

### 3.1.3    Iberia-Biscay-Ireland (*IBI*; IBI_ANALYSIS_FORECAST_PHYS_005_001)

*IBI* is developed by Mercator Ocean but is operated by the Spanish Port Authority (Puertos del Estado, Spain) and, although the model domain mainly corresponds to the northeastern Atlantic Ocean, the output simulation also covers the WMED to as far as the 5°E meridian (blue line in Fig. 1). The model grid is a subset of the Global 1/12° ORCA tripolar grid also used by the parent system (that provides initial and lateral boundary conditions) but refined to 1/36° horizontal resolution (∼2 km). The system is based on an eddy-resolving NEMO model application run at 1/36° horizontal resolution with 50 vertical z-levels. *IBI* does not include data assimilation; however a downscaling methodology is applied that improves the solution near to the open boundaries and the coasts (Sotillo et al., 2015; Aznar et al., 2016). Lateral open boundary data (temperature, salinity, velocities, and sea level) are interpolated from the daily *GLO* outputs (Sec. 3.1.1). These are complemented by 11 tidal harmonics (M2, S2, N2, K1, O1, Q1, M4, K2, P1, Mf, Mm) built from FES2004 (Lyard et al., 2006) and TPXO7.1 (Egbert and Erofeeva, 2002) tidal model solutions. River

runoff consists of a combination of daily observations (PREVIMER project), simulated data (SMHI E-HYPE model), a monthly climatology (GRDC), and the French hydrographic database known as "Banque Hydro" (http://hydro.eaufrance.fr). Topography is taken from the GEBCO08 dataset plus other local databases as reported by Maraldi et al. (2013).

## 3.2 Altimetry

The Mediterranean Sea gridded altimetry product from CMEMS is used to make an observational reference eddy track dataset. The daily sea level anomaly (SLA) along-track satellite observations are interpolated onto a $0.125° \times 0.125°$ grid. The spatial correlation length scales used for this regional product are set to $\sim$100 km, which is at the lower end of the range used for its global 0.25°counterpart ($\sim$90-150 km) (Pujol et al., 2016).

## 3.3 Argo

We use temperature and salinity profiles from Argo floats for a validation of the eddy composite results in Sec. 4.3. Argo floats are relatively sparse in the Alboran Sea, but over the 2013-2016 time period of this study their population across the Mediterranean did increase considerably (Sánchez-Roman et al., 2017). The Argo data were downloaded by *ftp* via the ECCO consortium website (ftp://ecco.jpl.nasa.gov/Version4/ Release3/profiles/) (Forget et al., 2015; Fukumori et al., 2017).

## 3.4 Eddy tracking

Version 3.0 of the *py-eddy-tracker*, a sea-surface-height based mesoscale eddy identification and tracking code developed by Mason et al. (2014), was applied to the daily SLA fields from altimetry and the three CMEMS models for the period 01/01/2013 to 30/06/2016. As the models provide the sea surface height (SSH), respective daily model SLA fields were obtained by taking the differences between daily model SSH and SSH means over the study period. The *py-eddy-tracker* uses an SSH-based contouring approach to eddy identification that is similar to the procedures described by Chelton et al. (2011b). The eddy tracker was configured to detect a wide range of eddy sizes and shapes. The same tuning parameters are used for each product, although the different grid resolutions (and relatively coarse correlation length scales used for *ALT*) between the products implies that the scales of detected eddy features will differ. The main implication is

that smaller eddies in the higher resolution models, *IBI* and to a lesser degree, *MFS*, may not be identified. As our main focus is the mesoscale eddies of the Alboran gyres we do not see this as a significant drawback to our experimental setup. The minimum and maximum parameter values were, for the effective radius $L_e$, 0.15°/1.5° and, for amplitude $A$, 0/150 cm. The shape error (Kurian et al., 2011) was 65%. The $L_e$ and $A$ parameters impose minimum and maximum eddy sizes, while the shape error (Kurian et al., 2011) excludes filaments and other elongated closed-contour structures that may not correspond to eddies. For the eddy tracking the minimum eddy lifetime was set to 5 days.

Time dependent outputs from the eddy tracker include eddy position, a speed-based (inner) radius ($L$) and an effective (outer) radius ($L_e$), amplitude ($A$), swirl speed ($U$) and eddy kinetic energy. Two useful ratios that can be obtained from these eddy properties are nonlinearity ($N = {}^U/_c$, where $c$ is the eddy propagation speed) and eddy intensity ($EI = {}^A/_L$). $N$ provides a measure of an eddy's capacity to trap fluid within its center; this occurs at values greater than unity (e.g., Chelton et al., 2011b). $EI$ is a potential proxy for the presence of elevated vertical motions (e.g., Frenger et al., 2015; Mason et al., 2017).

### 3.5 Eddy compositing

An eddy-centric composite analysis of the model prognostic variables was carried out following the procedures described by Mason et al. (2017). Briefly, for a given variable $p$ (e.g., temperature or salinity), we regrid the data at each vertical level to an eddy-centric grid where the coordinates span the range $\pm 4L$. An overview of this operation, from a regular grid in degrees to a spatial projection in kilometers and, finally, to the eddy-centric coordinate is provided in Fig. 2. This *normalisation* by the eddy radius of sequential fields permits us to then make multidimensional composite averages of the eddies and their properties.

Subregional composite averages are made by selecting only the eddies within certain predefined areas, such as those outlined in red in the Alboran Sea in Fig. 1. Further selection choices include selecting all the observations (i.e., we omit the tracking); selecting only observations from eddies that exceed a specified threshold lifetime; or we can select only those observations that are common (in terms of time and position, within some predefined ranges, and polarity) from the three models. Here we choose the second of these options, and use the same minimum lifetime threshold used for the eddy tracking (5 days; Sec. 3.4). We discarded the third option because the absence of data assimilation in *IBI* implies that this solution will drift

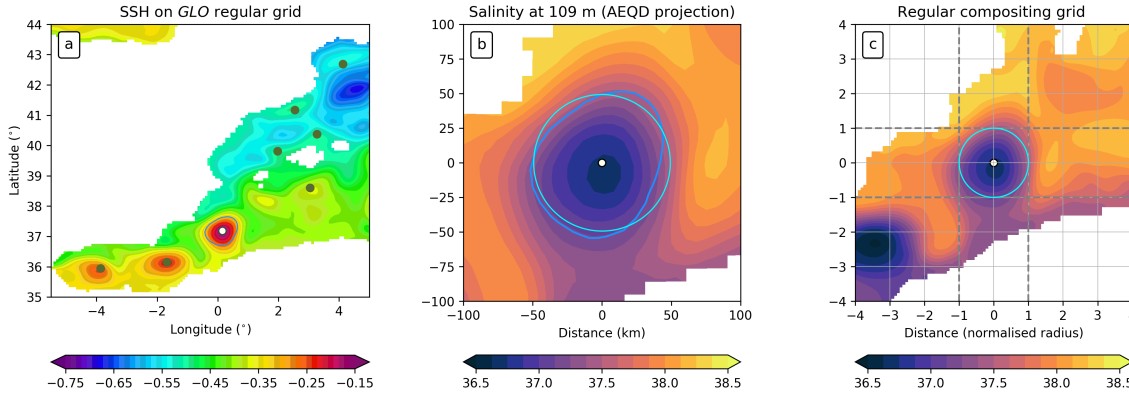

**Figure 2.** Illustration of the sequential compositing methodology using example SSH and salinity fields from *GLO* on 13/09/2013. (a) SSH is plotted on a regular grid over the western Mediterranean eddy tracking domain. The most intense of several eddy-tracker-identified anticyclones is marked in white near 0°E, 37°N and its corresponding speed-based contour is plotted in blue. Other identified eddies are marked in green. (b) A zoom over the intense *GLO* eddy of salinity at a depth of 109 m (plotted on an azimuthal equidistant projection; AEQD) shows the fresh salinity anomaly associated with the eddy. Light blue circle corresponds to the eddy radius. (c) The salinity field interpolated from the AEQD grid to the eddy-centric grid. Gray dashed lines highlight the radial extent of the eddy.

substantially from *GLO* and *MFS*, meaning that our overall sample size for compositing may be significantly reduced.

20    The core variables common to each model that were processed in the manner outlined above are potential temperature ($T$), salinity ($S$), SSH ($\eta$), and $u$ and $v$ velocity components. Model topography was available from *GLO*, and this was also interpolated to the eddy-centric grid. Anomalies of $T$ and $S$ in Sec. 4 are computed at each level and for every eddy instance by taking the difference between the original and a low-pass filtered field obtained from the convolution of a Gaussian kernel with a half width of 6° (e.g., Gaube

25    et al., 2014; Mason et al., 2017). Thus, $T^{'} = T - T_{\sigma=6}$, and similarly for $S^{'}$.

Finally, three additional external variables were similarly processed for eddy compositing:

- The normalised relative vorticity ($\varsigma/f$) is derived from $u$ and $v$, with $\zeta = \frac{\partial v}{\partial x} - \frac{\partial u}{\partial y}$ and $f$ the Coriolis frequency. The $\zeta$ operation is performed on the respective regular model grids, with $\partial x$ and $\partial y$ calculated using the haversine formula.

- Topography from the $1'$ Shuttle Radar Topography Mission (SRTM) dataset (Smith and Sandwell, 1997; Becker et al., 2009) is used as an intercomparison reference in Secs. 4.2.2–4.2.4.

- Mixed layer depths across the eddies were compiled by applying the density algorithm developed by Holte and Talley (2009) to the respective model $T$ and $S$ profiles. The range and seasonal variability of the eddy-centric MLD estimates is discussed in Sec. 4.2.5.

## 3.6 Eddy vertical tilt correction

We introduce a simple methodology to estimate eddy tilt. For every eddy observation, starting from the position of the eddy center at the surface level, we estimate the indices $i$, $j$ to the maximum of $|\varsigma/f|$ at each model depth level. No interpolation or vertical re-gridding is required. The indices provide (i) a means to estimate the distance at each level between the dynamical center of the eddy and its estimated surface position as determined by the eddy tracker, and (ii) the possibility to reconstruct each eddy variable (temperature, salinity, etc.) so that each level is aligned horizontally with the position of $|\varsigma/f|$. Illustrative figures of the impact on profile sections of $\zeta$, $T$ and $S$ are provided in Sec. A2; these can be compared with Figs. 7, 8 and 9 in Sec. 4.

## 4 Results

## 4.1 Eddy properties

Eddy property information covers data obtained directly from the eddy tracker, namely eddy position in time and space, radius, amplitude and swirl speed (e.g., Chelton et al., 2011b; Mason et al., 2014).

### 4.1.1 Eddy tracks

Tracks of detected eddies in the western Mediterranean indicate that the longest-lived eddies are typically found in the southern part of the WMED (Fig. 3). Anticyclones tend to be longer-lived than cyclones. The

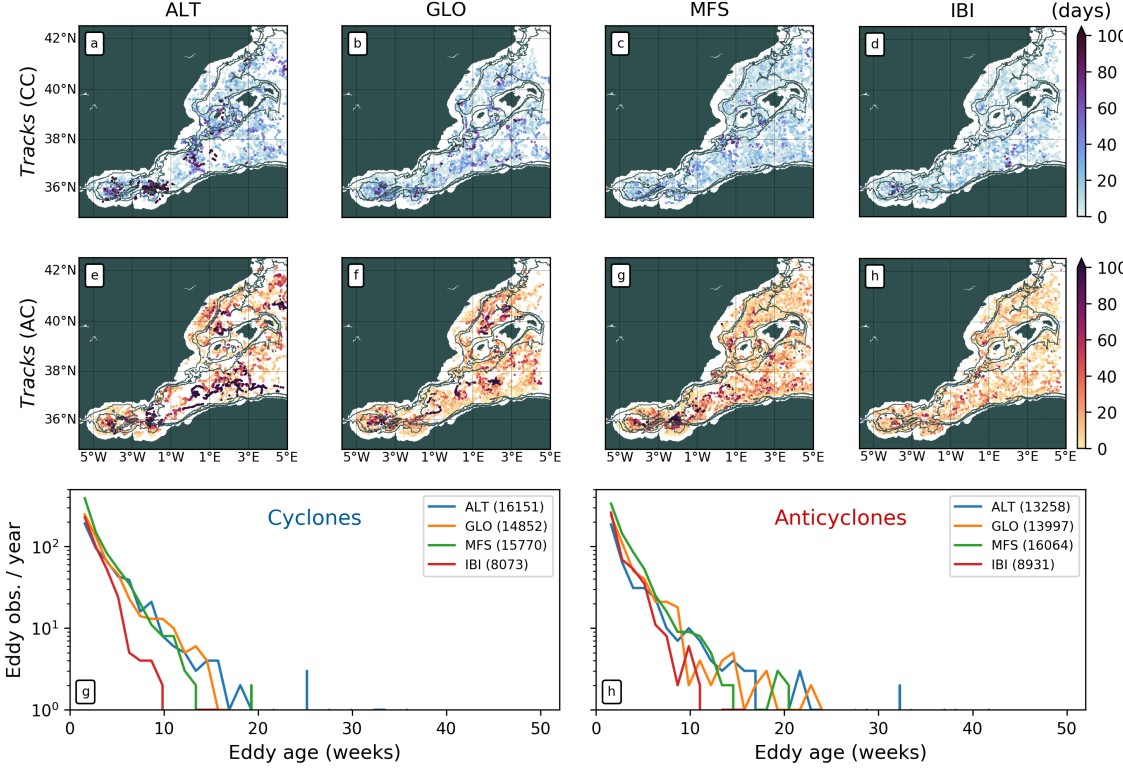

**Figure 3.** Scatter plots of (top row) cyclonic and (middle row) anticyclonic eddy observations in the western Mediterranean from altimetry and the three CMEMS models between 2013 and 2016. Colours indicate eddy age between 0 and 100+ days. Topographic contours from SRTM are plotted in gray at 500, 1000, 2000, 4000 m. Plots of eddy lifetimes of cyclones (bottom left) and anticyclones (bottom right) from altimetry and the three models. Numbers in the legends show the total numbers of eddy observations for each product.

quantity and duration of long-lived eddies is higher in *ALT* and *GLO*, and lower in *IBI*. The long-lived *ALT* eddies are concentrated inside the Alboran gyres and the Algerian Basin. There is also a small number of long-lived *ALT* anticyclones in the Balearic Sea. *MFS* and *GLO* have similar patterns of eddy distribution to *ALT*, but they have greater numbers of shorter-lived eddies (Fig. 3g,h). Large numbers of eddies are detected and tracked in *IBI* but they are of notably shorter duration than in the other products.

## 4.1.2 Eddy amplitude, radius and intensity

High levels of correspondence are found in eddy amplitude and radius distributions from the three models and altimetry over the western Mediterranean study region (Fig. 4). Maps of mean eddy amplitude show that larger amplitudes are found consistently across the southern regions (up to ~10 cm), and especially in the gyres of the Alboran Sea where amplitudes are ~10 cm (Fig. 4a-h). Anticyclones in the south tend to have larger amplitudes and are more prevalent than cyclones which, aside from *IBI*, are somewhat smaller outside of the Alboran Sea. The distributions of the amplitude patterns between the products are quite similar for anticyclones, but the cyclones in *IBI* (Fig. 4d) have noticeably higher amplitudes along the Algerian current axis than they do in the other products. In the Balearic Sea to the north, typical amplitude values are smaller at around 2.5 cm. There are only small differences between cyclones and anticyclones. Extreme amplitudes (>15 cm) are noticeable in the anticyclones of *ALT* and *GLO* at ~37°N, 0°E (Fig. 4e,f). This position corresponds to the Jason-1/Jason-2 satellite track (not shown) which could explain detection of a strong eddy amplitude signal in this region; *GLO* assimilates altimetry, as does *MFS* which also has a raised amplitude at ~0°E (Fig. 4g). Note also that the eddy identified in *GLO* in Fig. 2 occupies this same position, and a large eddy here is visible in the corresponding gridded altimetry map (not shown).

Cyclone and anticyclone radii are generally larger in the southern parts of the study domain than they are in the north (Fig. 4i-p). The patterns between the models and altimetry are very similar for the anticyclones, aside from the Balearic Sea in *ALT* where the radii are slightly larger. There is more variability in the cyclones, where *GLO* has noticeably smaller radii in the Alboran gyres. *ALT* and *IBI* cyclone radii are marginally larger than their counterparts in *GLO* and *MFS*. Differences in radii for eddies of both signs between *ALT* and the models are most apparent in the Balearic Sea, where *ALT* eddies are consistently larger.

Eddy intensity maps from all products show progressive increases with resolution in both cyclones and anticyclones (Fig. 5a-h). *EI* is consistently at or above 0.2 cm km$^{-1}$ within the Alboran Sea. In *ALT* the western and eastern Alboran gyres are clearly distinguishable in the *EI* signal. *ALT* cyclones along the Algerian coast to the east have weak *EI*, whereas anticyclones have some of the largest values. Examination of Fig. 4a,e indicates this variability is largely determined by eddy amplitude rather than radius.

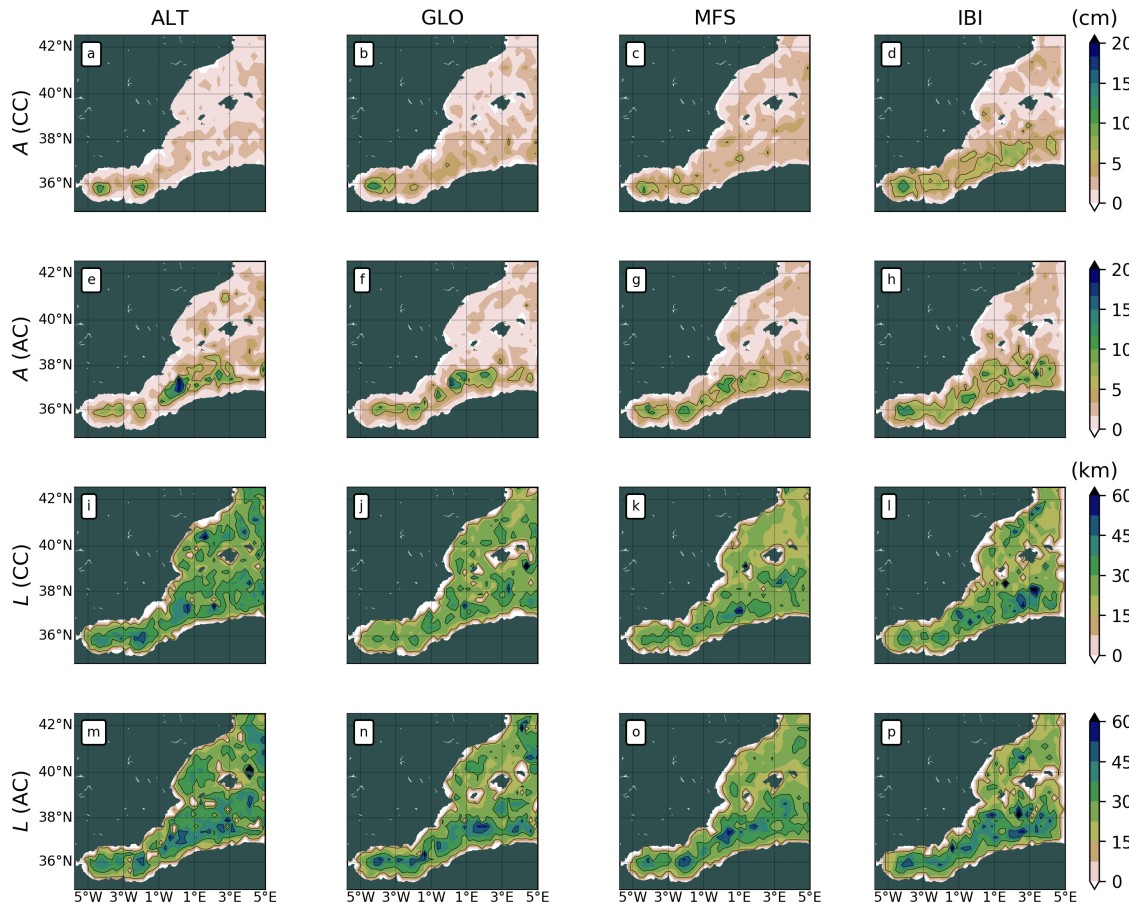

**Figure 4.** Maps of mean eddy amplitude (a-h) and radius (i-p) over the western Mediterranean. Columns for *ALT*, *GLO*, *MFS* and *IBI*. Top and bottom paired rows for cyclones (CC) and anticyclones (AC).

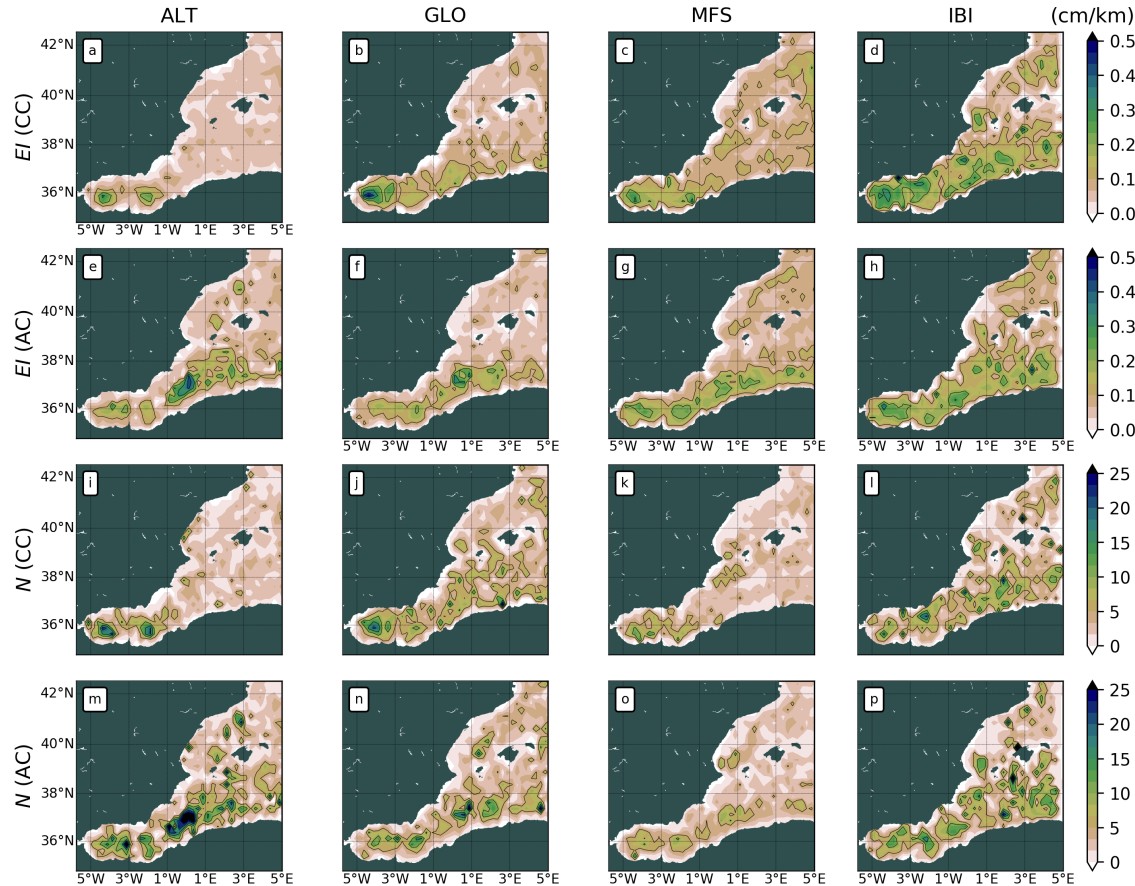

**Figure 5.** Maps of mean eddy intensity (a-h) and nonlinearity (i-p) over the western Mediterranean. Columns for *ALT*, *GLO*, *MFS* and *IBI*. Paired rows for cyclones (CC) and anticyclones (AC). Maps of the numerator ($U$) and denominator ($c$) for nonlinearity in i through p are shown in Sec. A1.

**Table 2.** Subregional eddy counts, and mean and median coordinates and properties from the eddy tracker for anticyclones in the western Alboran gyre (WAG), eastern Alboran gyre (EAG), and Cartagena frontal region (CRT). $\bar{\cdot}$ and $\tilde{\cdot}$ denote the respective means and medians of eddy positional coordinates (degrees) and radii (km) in each subregion defined in Fig. 1.

| Subregion | Model | $N$ | $\overline{Lon}$ | $\overline{Lat}$ | $\widetilde{Lon}$ | $\widetilde{Lat}$ | $\bar{L}$ | $\widetilde{L}$ |
|---|---|---|---|---|---|---|---|---|
| WAG | GLO | 1231 | -4.11 | 36.02 | -4.06 | 36.03 | 33.2 | 29.2 |
| | MFS | 1374 | -4.12 | 35.90 | -4.18 | 35.85 | 30.6 | 29.5 |
| | IBI | 1048 | -4.24 | 35.84 | -4.26 | 35.74 | 28.9 | 25.3 |
| | ALT | 976 | -4.1 | 35.87 | -4.1 | 35.84 | 32.9 | 33.1 |
| EAG | GLO | 1033 | -2.34 | 35.90 | -2.27 | 35.86 | 34.5 | 32.0 |
| | MFS | 1235 | -2.23 | 35.92 | -2.17 | 35.91 | 33.3 | 31.1 |
| | IBI | 845 | -2.36 | 35.92 | -2.26 | 35.91 | 33.9 | 31.8 |
| | ALT | 862 | -2.27 | 35.90 | -2.10 | 35.91 | 33.8 | 34.3 |
| CRT | GLO | 1040 | -0.50 | 36.66 | -0.43 | 36.64 | 31.6 | 28.9 |
| | MFS | 1024 | -0.67 | 36.64 | -0.71 | 36.67 | 35.4 | 26.0 |
| | IBI | 764 | -0.66 | 36.68 | -0.68 | 36.65 | 35.4 | 32.8 |
| | ALT | 644 | -0.60 | 36.70 | -0.56 | 36.80 | 33.7 | 33.9 |

## 4.2 Subregional eddy composites

In this section we focus on the anticyclones in the western and eastern Alboran gyres (WAG and EAG) and the Cartagena frontal region (CRT) to the east. The three subregions are outlined in red in Fig. 1. As anticyclones are the dominant signal in the Alboran Sea we omit cyclones from our analysis. (We do however provide the cyclonic counterparts to the vertical section figures in this section in Sec. A4.) Horizontal and vertical subregional eddy composites illustrate the variability in vorticity, temperature and salinity across the gyres of the Alboran Sea. A summary of the eddy properties from the eddy tracker for these subregions is provided in Tab. 2.

### 4.2.1 Horizontal subregional eddy composites

Horizontal anticyclonic eddy composites of $\varsigma/f$, $T'$ and $S'$ in Fig. 6 reveal the intensity of the eddy property anomalies of the three defined subregions of the Alboran Sea. Each variable is plotted at the median positions of the eddy coordinates that contribute to each subregion. The eddy positions between the models are very similar. In the WAG, the median eddy positions for each model are located in the center of the gyre, which corresponds to the deepest water. *GLO* is found north of the Vizconde de Eza seamount (located at $\sim$4°W, 35.8°N in Fig. 1), while *MFS* and *IBI* are to its west. In the EAG the eddy positions, lying over the southern 1000 m isobath of the eastern Alboran basin, are virtually indistinguishable. The CRT coordinates are located about 0.75° further north in the broad deep water depression that opens into the Algerian basin with, again, little observable difference in median eddy position between the models. The plotting depth in Fig. 6 of each variable corresponds to the respective absolute maximum in the water column at the eddy centre. These depths vary, being very shallow for $\varsigma$, some tens of meters deeper for $T'$, and generally below 100 m for $S'$. The anomalies tend to be approximately confined within the limits of the two radius estimates, $L$ and $L_e$. The anomalies are plotted out to a radial extent of $2L$, with the region between $L$ and $2L$ defined by Frenger et al. (2015) as the *eddy impact area*[1].

The near surface negative vorticity values inside the $L$ and $L_e$ radii are variable according to both subregion and model in Fig. 6 (top row). For each model, $|\varsigma|$ intensity in each subregion successively decreases from the west (WAG) to east (CRT). There is also an overall increase in $|\varsigma|$ in each subregion between the models; *GLO* is weakest and *IBI* strongest. The model increases can be explained by the increasing model resolution. In the WAG, *IBI* $\varsigma/f$ approaches 1 indicating the possible admittance of ageostrophic motions; however, the incoming Atlantic Jet in *IBI* is suspected to be too strong such that these $\varsigma$ values may be an overestimate (Ruiz et al., 2018). Outside of the eddy radii (i.e., from approximately $L$ to $2L$) the $\varsigma$ values are uniformly positive.

The temperature anomalies in the middle row of Fig. 6 vary in depth between $\sim$20 and 155 m. There is inconsistency between the models in terms of the depths in each subregion. For example, the shallowest anomaly in *MFS* is in the WAG, whereas *IBI* has its *deepest* anomaly in this subregion. The most intense values of $T'$ are found in *GLO* in the WAG. In the eddy impact region of this eddy composite there is intense

---

[1]The eddy impact area as defined by Frenger et al. is the area between one and three times the eddy radius around the eddy center; here, given the size of the Alboran gyres and the confined Alboran domain, we choose to use twice the radius.

negative $T'$ in the southern and western quadrants; on the eastern side there is an abrupt transition to an arc
of positive $T'$ over the Alboran Trough at $\sim$3.5°W. The *MFS* WAG pattern is similar although the strength
of the anomaly is smaller. *IBI* has a strong $T'$ core, with a nearly continuous negative $T'$ in the eddy impact
region. The EAG and CTR $T'$ patterns are quite consistent between the models, although the *IBI* CRT $T'$ is
noticeably stronger than in the other models.

The salinity anomalies in the bottom row of Fig. 6 are the most consistent of the three variables. The
deepest anomalies are in the WAG, and the shallowest in the CRT. The fresh $S'$ composites in the eddy
cores have similar magnitudes, and the same for the opposite sign in the eddy impact region. In contrast to
temperature, the same sign $S'$ in the impact region is found both south and north of the eddy cores. The most
intense $S'$ is in the WAG in *IBI* at a depth of 155 m; this contrasts with *GLO* and *MFS* minima at 130 and
123 m, respectively.

### 4.2.2 Vertical eddy composite: the Western Alboran Gyre

Vertical sections through zonal and meridional anticyclonic WAG eddy composites in Fig. 7 reveal good
structural agreement between the models in $\varsigma/f$, $T'$ and $S'$. There are also some striking differences. The
widths of the anomalies broadly correspond to the mean radii marked in orange in Fig. 7 for each eddy
composite (Tab. 2). The *tilt* of the eddies is described by the vertical brown lines; tilt is estimated based
on the position of absolute maximum $\zeta$ within the eddy radius at each vertical $z$-level for the respective
models. (See Sec. 3.6 and Sec. A2.) There is good agreement in eddy tilt between the models, especially in
the meridional composites.

Concerning the eddy composite topography, the zonal plots indicate shoaling towards the Strait of Gibral-
tar in the west. The meridional plots show the eddies to be centered over the 200 m deep trough just to the
north of the Alboran Ridge (see inset in Fig. 1) at the bottom of the Alboran basin. Owing to the succes-
sively higher model grid resolutions, more topographic details are visible in *IBI* and *MFS* than in *GLO*. The
median *IBI* WAG position is to the west and south of the *MFS* and *GLO* positions; this explains the apparent
shallower *IBI* topographic composite.

The intensity of surface-intensified negative $\varsigma/f$ more than doubles between *GLO* and *IBI*, and the anoma-
lies extend downward to between $\sim$150 m (*GLO*) and $\sim$200 m (*IBI*) (top row Fig. 7).

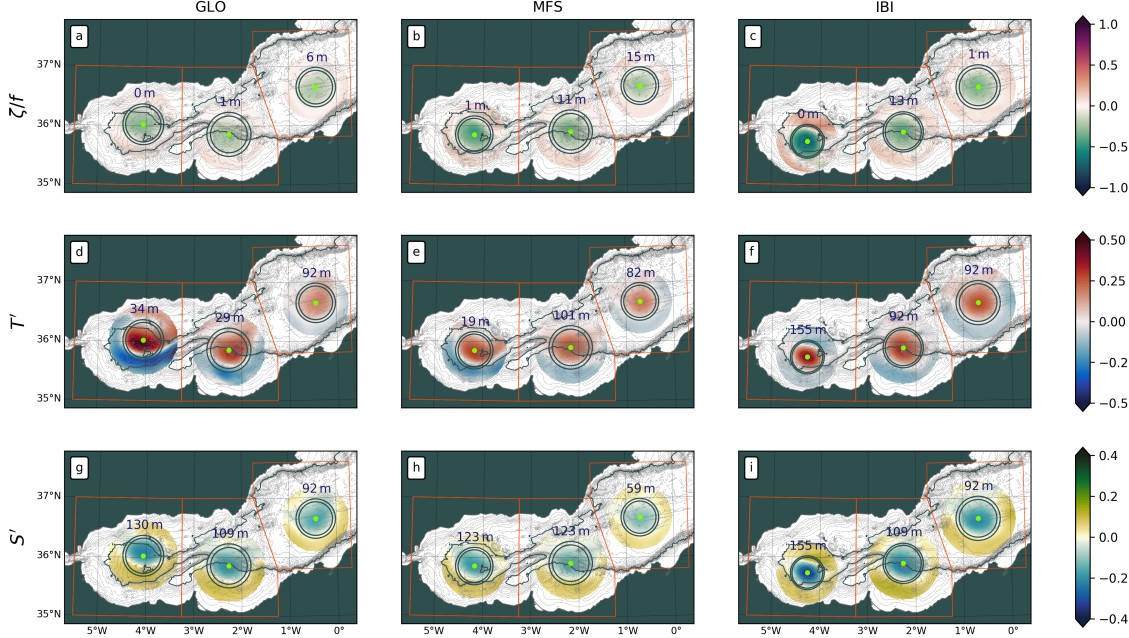

**Figure 6.** Anticyclonic eddy composites of $\varsigma/f$ (top row), $T'$ (middle row), and $S'$ (bottom row) in three subregions in the Alboran Sea from the three CMEMS models, *GLO* (left column), *MFS* (middle column) and *IBI* (right column). Orange boxes indicate the bounds of each subregion used for the compositing. The mean position of each eddy composite is shown by neon green dots; the depth in meters at which each composite is plotted corresponds to absolute maximum of the variable over the water column at the position of the eddy center. Gray circles around each dot correspond to the mean speed and effective eddy radii. SRTM topographic contours are plotted in gray from the surface to the bottom at intervals of 100 m; the 1000 m isobath is plotted in dark gray.

*GLO* and *MFS* $T'$ sections are quite similar with two positive cores, one between the surface and about 50 m, and the other at 110 m (middle row Fig. 7). The *IBI* section is distinct as it has just one core at 150 m, and no expression at the surface. The zonal sections have weak negative $T'$ anomalies on their eastern flanks in the eddy impact area beyond the radius. The strongest $T'$ anomalies are seen in *GLO* (upper core) and *IBI*.

WAG $S'$ from the models have similar structure in Fig. 7 (bottom row). Single cores of negative $S'$ are centered at around 135 m. The upper surfaces of these anomalies shoal towards the north and east, producing a small surface expression within the northeast quadrant of each eddy.

### 4.2.3   Vertical eddy composite: the Eastern Alboran Gyre

Structural agreement similar to that of the western Alboran gyre above is also visible in the vertical sections of $\varsigma/f$, $T'$ and $S'$ in the eastern Alboran gyre (Fig. 8). The eddies here have marginally larger radii than in the WAG (Tab. 2), and lie in slightly deeper waters. The intensity of $\varsigma/f$ progressively increases from *GLO* to *IBI*. Zonal sections of $\varsigma/f$ are symmetric with weak tilt (top row Fig. 8); the eddies are centered over topography that descends towards the east. The meridional sections of $\varsigma/f$ are asymmetric with strong agreement between each model. The eddies tilt towards the north over the first $\sim$100-175 m, then back towards the south down to $\sim$600-700 m where they begin to feel the topography on their southern flanks.

Both anomalies $T'$ and $S'$ in the EAG are slightly weaker than those of the WAG. In contrast to the WAG, zonal $T'$ in *GLO* and *IBI* has a surface signature comparable to that below; *MFS* on the other hand has a weak surface signature that only becomes significant at about 30 m. The zonal $S'$ structure is the reverse of $T'$, in *GLO* and *IBI* the anomaly is centered between $\sim$ 50 and 150 m whereas in *MFS* it reaches the surface. The meridional $S'$ sections emphasise the strong northward tilt of the eddies. Negative $S'$ is concentrated north of the eddy center between the surface and $\sim$100 m in each of the models.

### 4.2.4   Vertical eddy composite: the Cartagena Frontal Region

Detected eddies in the Cartagena Frontal region to the east of the EAG have weaker tracer and circulation anomalies than in the Alboran gyres (Fig. 9). Eddy radii here from *GLO* and *MFS* are slightly smaller than in the WAG and EAG; the *IBI* CRT eddies meanwhile have the largest radii for all three subregions (Tab. 2). Depths in the CRT reach 2500 m and seafloor gradients are smaller than those in the Alboran gyres of Figs. 7 and 8. The same $\varsigma/f$ intensity increase from *GLO* to *IBI* observed in the WAG and EAG is visible in Fig. 9

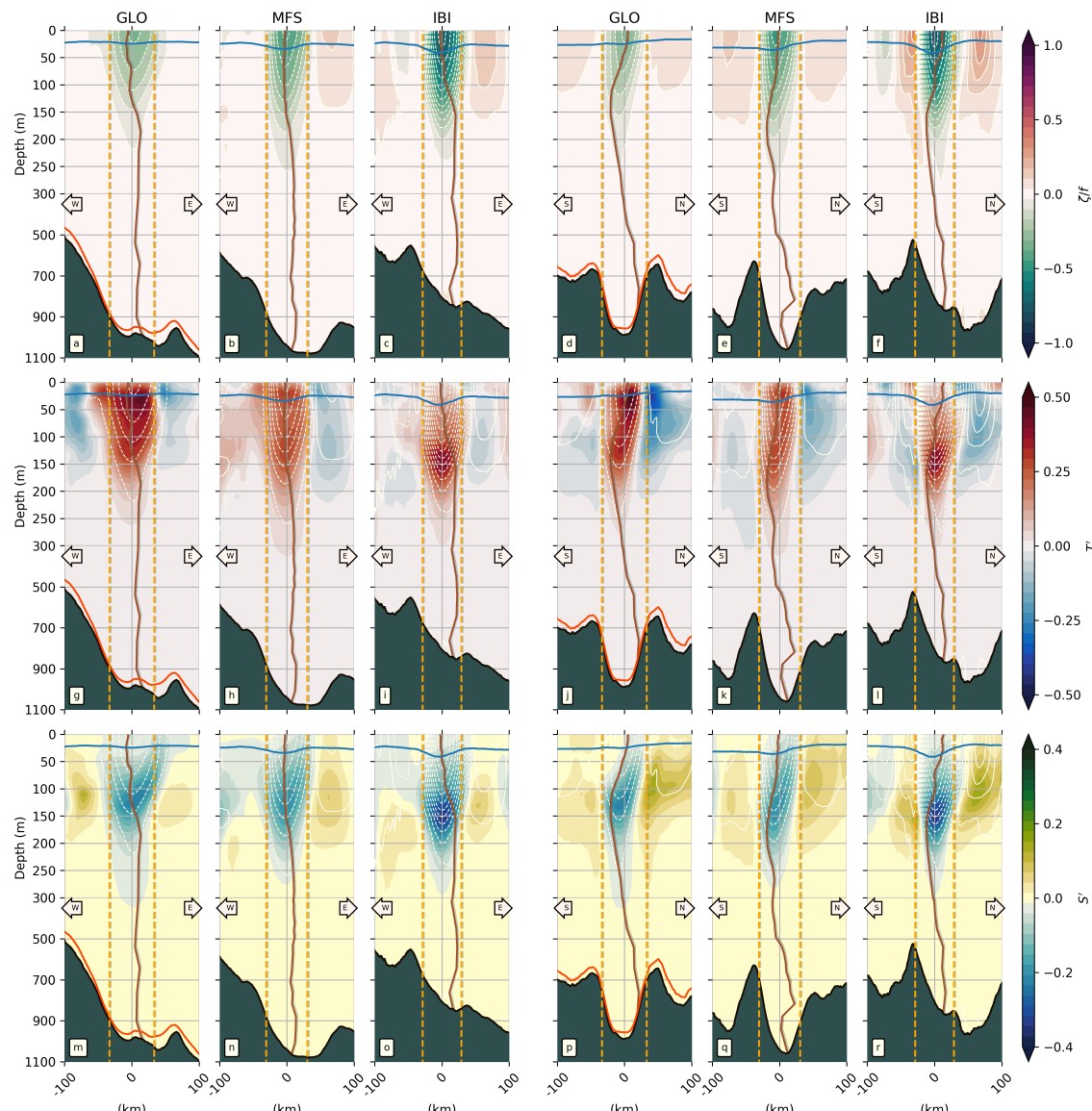

**Figure 7.** Eddy composite sections from *GLO*, *MFS* and *IBI* in the western Alboran gyre. Left-hand-side (right-hand-side) columns show zonal (meridional) sections of (top to bottom) relative vorticity, temperature anomalies and salinity anomalies, from the surface to the ocean floor. The central position of each section is the median of the longitudes and latitudes associated with the eddy observations used to make the composites (Tab. 2). Blue lines indicate the mixed layer depth; the dotted blue line corresponds to the MLD from the *MFS* model. The vertical brown line in each section is the vorticity-based tilt correction (see Sec. 3.6). Vertical orange dashed lines indicate the boundaries of each composite eddy based on its mean radius estimate from Tab. 2. Composite topographic profiles in black from SRTM, and also in red from *GLO*. Note change of vertical scale at 300 m.

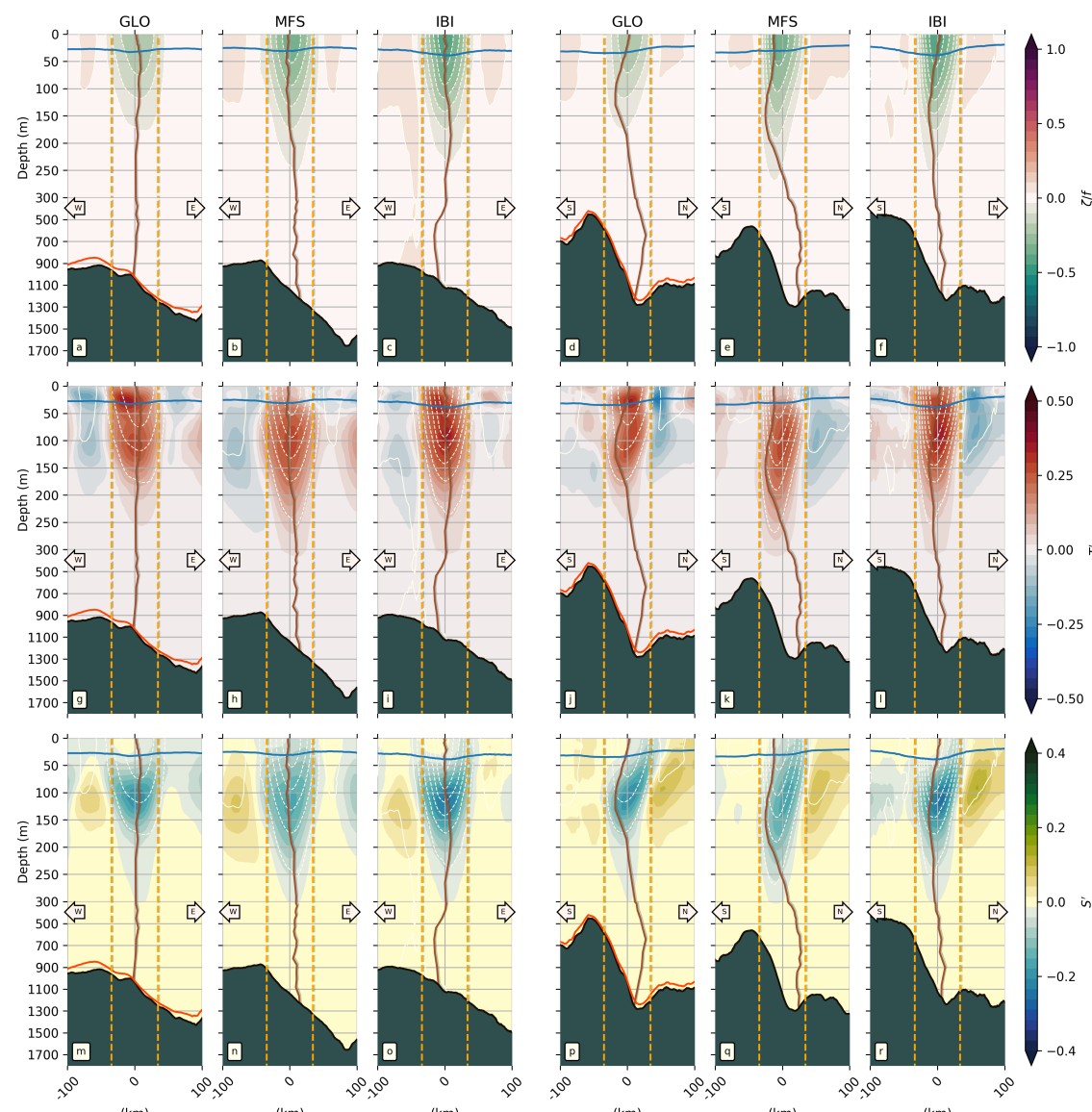

**Figure 8.** Same as Fig. 7 but for the eastern Alboran gyre.

(top row). *MFS* and *IBI* $\varsigma/f$ zonal sections are symmetric with small tilt; *GLO* has pronounced westward tilt down to about 150 m. The meridional $\varsigma/f$ sections are again asymmetric, with agreement between the models. The eddies tilt northward between the surface and ~125 m. The eddies are centered over the deepest isobath in both the zonal and meridional directions. Tracer anomalies $T'$ and $S'$ in the CRT are weaker than in the Alboran gyres. The cores with maximum $T'$ are found at around 100 m depth in the three models, with *GLO* slightly deeper (Fig. 9j) and *MFS* shallower (Fig. 9k). In the upper ~30 m $T'$ is near zero in *GLO* and *MFS*, while *IBI* has a weak positive $T'$ between the surface and ~20 m. The vertical extents of the $T'$ anomalies progress from around 250 m (*GLO*) to 800 m (*IBI*). The anomalies are compensated by negative $S'$ anomalies of broadly similar structure (Fig. 9m-r).

### 4.2.5  Seasonal mixed layer depth

Good agreement in the seasonal cycle of the WAG mixed layer depth between the models is evident in Fig. 10. The most interesting observation from the model estimates is the large variability, both seasonal as well as intra-seasonal, with the latter prominent in winter and autumn. As shortwave solar radiation increases from spring to summer, it induces strong stratification that is maximum at the end of the summer, leading to shallow MLDs of between ~5 and 15 m. In autumn, with decreasing shortwave radiation forcing, the MLD deepens to maxima of around 50 m (75 m for *IBI*). In winter, under the influence of wind-forced mixing processes, the model MLD estimates in the WAG are the deepest over the seasonal cycle, with *IBI* again having the maximum values with depths below 100 m. Our MLD estimates from the model outputs are in good agreement (especially in summer and autumn) with climatological Mediterranean MLD values reported by Houpert et al. (2015) that are based on density computed from observations. Small differences can be explained by (i) different methodological approaches (here we use the density algorithm of Holte and Talley, 2009) and, (ii) that our estimates are, by design, biased in that we only sample eddies.

The large variability in MLD over the course of a year may have important implications for upper layer processes, such as vertical motions associated with mesoscale structures (i.e., the WAG, and EAG) that promote exchange of mass, heat and tracers between the surface and the ocean interior. Seasonal MLD variability for the EAG and CRT subregions is similar to that shown for the WAG in Fig. 10; see Figs. S18 and S19 in Sec. A5.

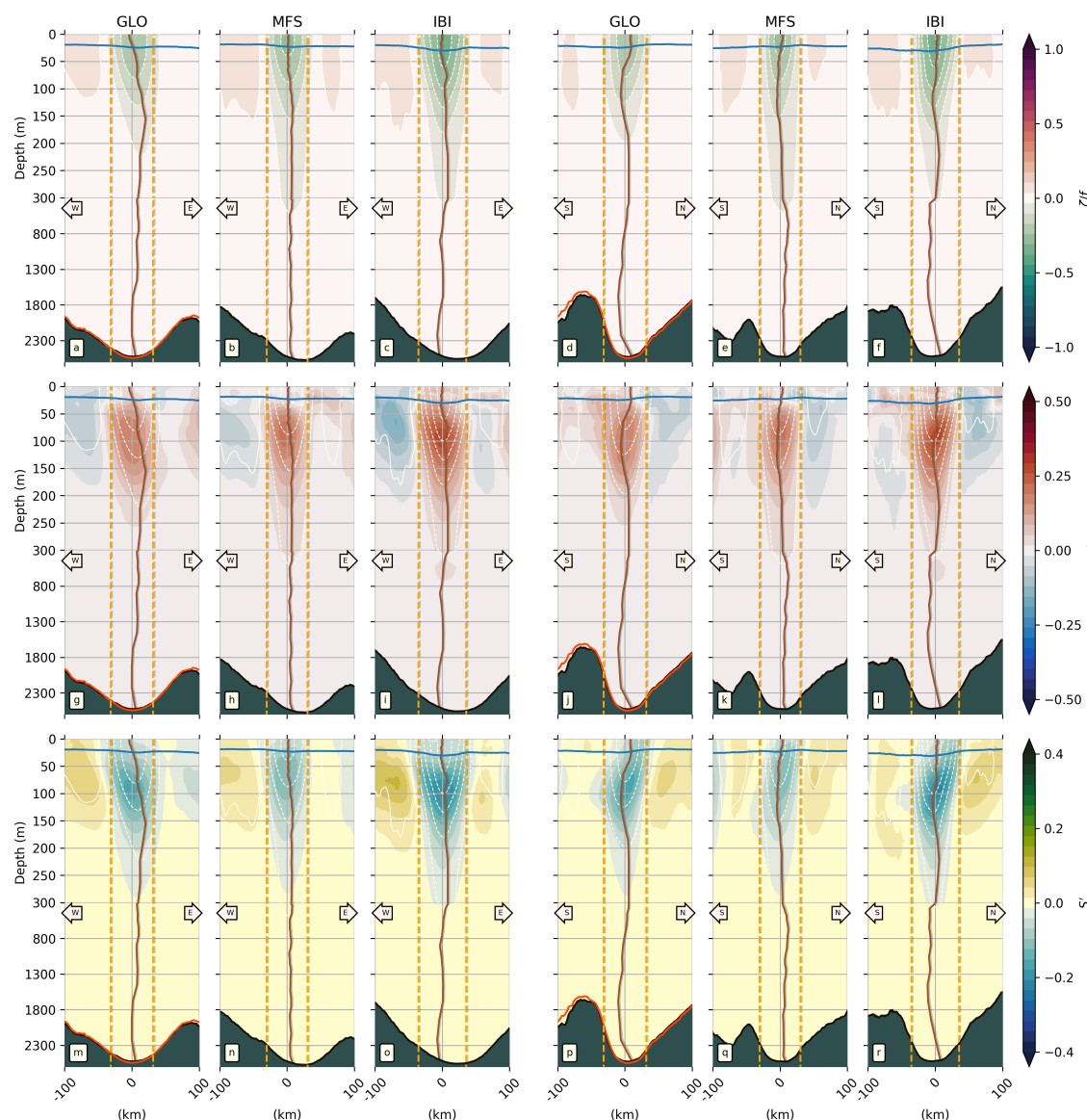

**Figure 9.** Same as Fig. 7 but for the Cartagena frontal region.

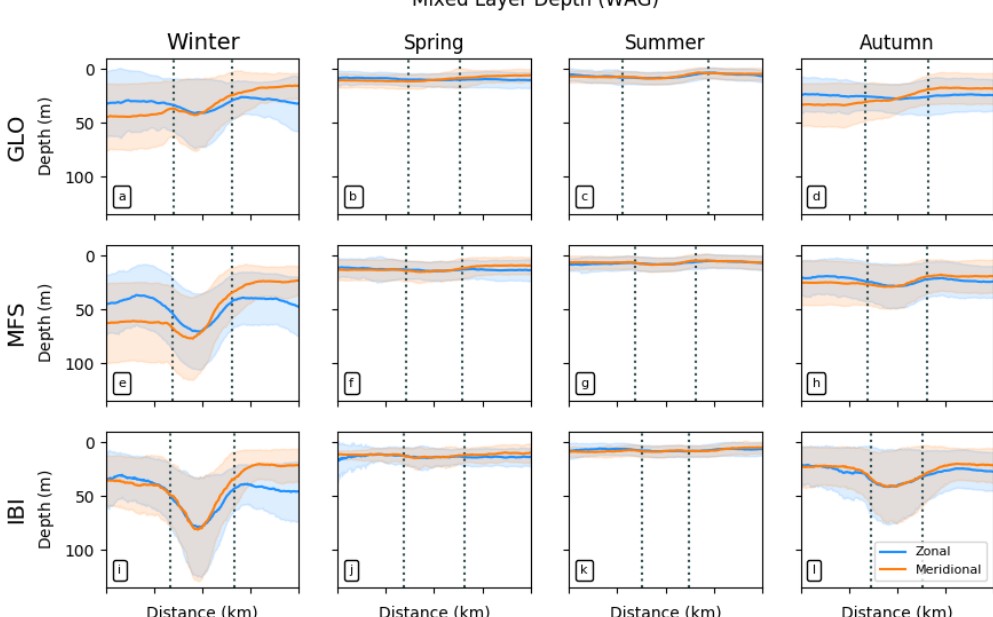

**Figure 10.** Mean and standard deviation of the seasonal mixed layer depth in anticyclones in the western Alboran gyre for *GLO* (top row), *MFS* (middle row) and *IBI* (bottom column). Zonal (meridional) profiles in blue (orange). Vertical dotted lines mark the mean eddy radius from the center.

## 4.3 Validation with Argo data

An important final point concerns validation of the above eddy composite results. Our present operational *in situ* observing system in the WMED is not able to provide sufficient density and frequency of observations for the creation of a comprehensive reference eddy composite dataset. This is especially the case in the semi-enclosed Alboran Sea where instrument residence times, Argo floats for instance, are very short owing to the large density gradients and associated strong currents. The current best approach is to use the CMEMS ARMOR3D product (as done by Mason et al., 2017, in the Brazil-Malvinas Confluence). Here we cannot use ARMOR3D because (i) it is not presently recommended for the Mediterranean Sea (S. Mulet, pers. comm.)

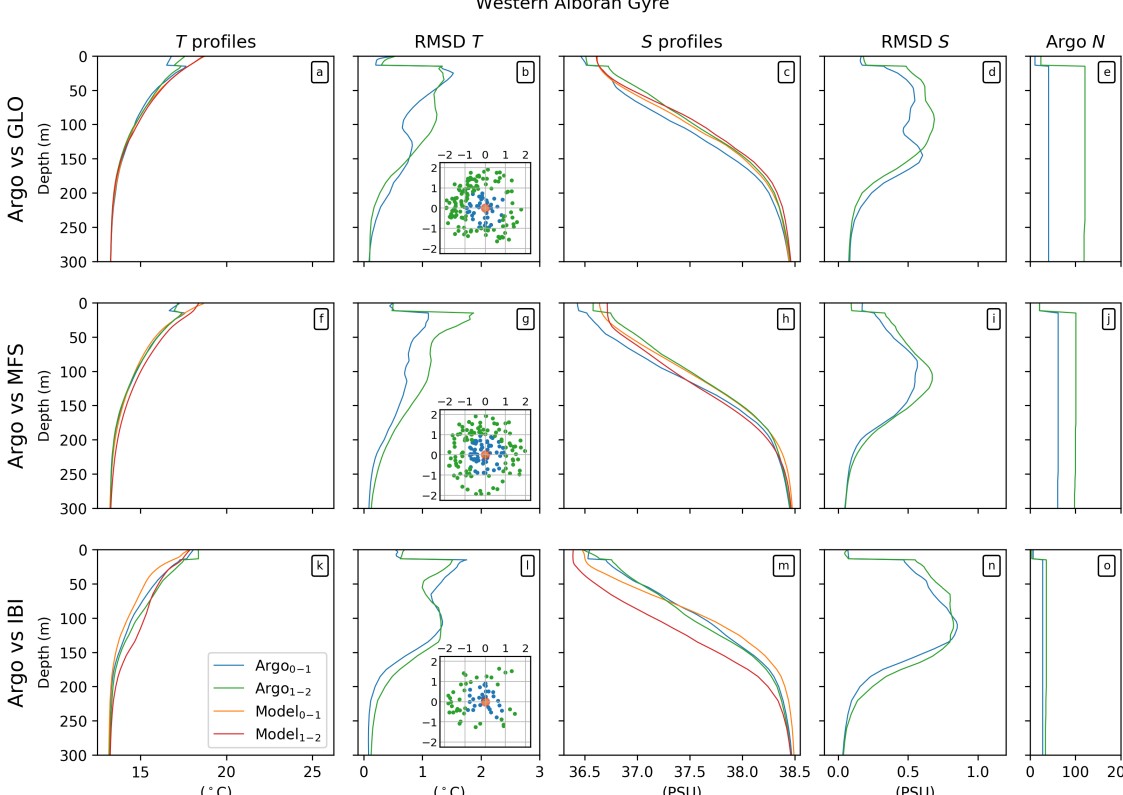

**Figure 11.** Comparison of temperature and salinity depth profiles at the center of each model anticyclonic eddy and its nearest Argo observation in the western Alboran gyre subregion. The first column shows mean $T$ profiles for the respective models and Argo; the second column shows the corresponding $T$ root mean square deviations with depth between the model and Argo $T$ observations. Argo profiles coloured blue (green) are located inside (outside) the eddy radius, viz. $0$-$1L$, $1L$-$2L$, as indicated in the inset eddy-centric-coordinate maps in column two that show the relative (to the eddy center) positions of each Argo profile. The respective model profiles corresponding to the Argo selections are coloured orange and red. Profiles in columns three and four show the results for $S$. The Argo sample size $N$ at each depth for each model is found in column five. Rows one to three show *GLO* (top), *MFS* (middle) and *IBI* (bottom).

and, (ii) its current 0.25° grid resolution is too coarse for meaningful representation of the semi-enclosed Alboran Sea. Despite the limitations noted above, our only resource therefore for validation is individual $T$ and $S$ profiles from Argo corresponding to 2013-2016 study period. We computed the root-mean-square deviations (RMSD) between the model $T$ and $S$ profiles at the center of each anticyclonic eddy, and two cohorts of Argo profiles; the first cohort is composed of all the profiles within the eddy radius (0-1$L$), and the second those in the eddy impact region (1$L$-2$L$). For both cohorts, the date of each profile is required to correspond to that of the model eddy observation. The expectation is that results from the first ARGO cohort, which are nearer to the eddy center than the second cohort, will produce a smaller RMSD. This is generally what we find in all three Alboran Sea subregions: the WAG in Fig. 11, and the EAG and CRT in Figs. S20 and S21. Across the Alboran Sea, the positive impact of data assimilation in *GLO* and *MFS* is readily apparent in comparison with *IBI*. We can therefore state with some confidence that the eddy composite results in Figs. 6 through 9 are reasonably accurate. Furthermore, the consistency between the models in vertical eddy structure for each variable, in both $x$ and $y$ planes, provides further reason for confidence. (More examples of this consistency can be seen in five further subregions across the WMED in Secs. A3 and A4.) Nevertheless, a higher resolution version of ARMOR3D for the Mediterranean (and other regional seas) much like, for example, the 1/8° Mediterranean altimetric SLA and ADT CMEMS products, would be a useful resource for model validations such as presented here.

## 5   Discussion

An eddy tracker and eddy-centric compositing techniques have been applied to the outputs of three CMEMS model products in order to assess their ability to reproduce mesoscale variability and three dimensional structure (eddies). Our results suggest that this approach can yield useful feedback to the developers of the CMEMS operational models, while also enabling regional and subregional characterisations of three-dimensional mesoscale eddy structure that is of interest to users of the CMEMS products.

It is important however to stress that we do not see the techniques used here as a substitute for other existing means of model evaluation and validation. Numerous sensitivity tests may be required to identify the sources of minor differences between runs. For example, the disparities in vertical eddy structure in Sec. 4.2 may arise from differences in the vertical discretisation, or from the mixing scheme, or from di-

vergence resulting from the advection scheme, or the projection to depth of surface information by the data assimilation.

## 5.1 Impact on future MFC product improvement

Beyond the added value of simply having an additional diagnostic to validate and compare different CMEMS products, we find that an important benefit of the eddy-centric composite information is the potential for insight into the choices made for the model configurations. The eddy tracker enables evaluation of the different numerical choices made when designing a model setup. For example, the horizontal advection scheme and its associated dissipation is strongly linked with eddy lifetime. The lateral boundary conditions (free-slip versus no-slip) at the coast modify horizontal current shear, hence leading to generation of more, or less, eddies. A second example concerns the possibility, based on the eddy composite information, of tuning of the vertical advection schemes in the models. Regarding the CMEMS reanalyses and forecasts, metrics from an eddy tracker may aid in the diagnoses of the work done by the data assimilation schemes, and their capability to accurately reproduce mesoscale structures in space and time. An eddy tracker can also be helpful in assessing and improving the vertical projection of surface information that is done by the data assimilation schemes, or to validate the gradual incorporation of the increments computed by the data assimilation schemes.

The benefit of data assimilation in *GLO* and *MFS* is illustrated in Sec. 4.3 where RMSDs are computed for Argo profiles of $T$ and $S$ inside and just outside the periphery of individual eddy observations, and corresponding model profiles at the center of the eddy. *GLO* and *MFS* are found to have smaller RMSD profiles than *IBI* for both tracer variables, which we can reasonably expect given the data assimilation.

The value of including tides is less easy to demonstrate. Tides are generally weak in the Mediterranean Sea as a whole, but they are relevant in the Strait of Gibraltar (e.g., Candela et al., 1990). Owing to their effect on the thermohaline circulation of the Mediterranean Sea, namely upper- and intermediate-layer cooling and saltening, many have argued for their inclusion in Mediterranean Sea model configurations (e.g., Harzallah et al., 2014; Naranjo et al., 2014). In the WAG in Fig. 7 we showed that *IBI*, that has tides, has a very small positive $T'$ in the upper 75 m in comparison with *GLO* and *MFS* which both have stronger positive $T'$. This difference may simply be a result of the absence of assimilation in *IBI*. But there remains the possibility that this discrepancy in the respective $T'$ differences in Fig. 7 is at least partially the inclusion of tides in *IBI*;

*GLO* and *MFS* lack the tidal-induced vertical mixing across the WAG that acts to cool the surface mixed layer. If this suggestion can be confirmed by the respective MFC engineers then a recommendation might be made to include tides in future versions of *GLO* and *MFS*. However, inclusion of tides may not be a trivial task in the case of a global model such as *GLO*, and so extensive cost/benefit research should be carried out beforehand. Other factors that could lead to discrepancies like the $T'$ differences we have observed include the different bulk heat flux parameterisations used at the surface; turbulence closure schemes and vertical mixing parameterisations; and numbers of vertical levels and their distribution. These are all aspects that the MFC engineers will take into consideration.

Model resolution appears to have an impact on the strength of the eddy properties and their ratios in Sec. 4.1, and the $T$, $S$ and $\zeta$ composite anomalies in Sec. 4.2. The differences become apparent by comparison with the figures in Sec. A2 where the tilt correction is applied during the making of the composites. Notice the larger numbers of $\zeta$ contours (white) in Figs. S2-S4, and also the negative $\zeta$ that extends all the way to the seabed in each of the models. Choice of model resolution is highly dependent on the size of the domain to be used; for the moment it is unlikely that in the near future we will see the global *GLO* at the resolution of *IBI*. On the other hand, the new version of *MFS* that was released during the writing of this paper in 2018 has a horizontal resolution of $1/24°$ (a 33% increase over the version used here), and double the number of vertical levels.

## 5.2 Potential benefits for CMEMS users

There is increasing interest and demand in the CMEMS user community for information and data on mesoscale structures (e.g., Crosnier and Delamarche, 2019). The results presented here suggest that developing an operational version of these techniques for eddy tracking, and eddy-centric compilation of a range of diagnostic variables, could be a useful addition to the CMEMS catalogue. These can be considered as *novel diagnostics*, and could contribute to a training database of mesoscale features that could be exploited for the purposes of machine learning. In the near future it is expected that CMEMS and/or downstream users will be able to access automated procedures that can detect mesoscale patterns in the ocean.

## 6   Conclusions

New insight is provided into the mesoscale content of three CMEMS operational model products, *GLO*, *MFS* and *IBI*, using a robust, sea-surface-height-based eddy identification and tracking tool. The analysis period is 2013 through 2016. Maps for each model product of mean eddy properties, including position, lifetime, radius and amplitude, reveal general consistency over the western Mediterranean Sea study region. The models that include data assimilation, *GLO* and *MFS*, approximate most closely the eddy property distributions observed with contemporaneous SSH observations from altimetry. Knowledge of eddy location enables construction of subregional 3D eddy composites of the model prognostic variables such as temperature, salinity and relative vorticity. Eddy centric composites of these variables in three subregions of the Alboran Sea reveal the strong frontal characteristics associated with the Alboran gyres. The eddy centric composites also provide feedback about the impacts of inclusion of data assimilation, tides or other parameterisations in the respective model configurations. The positive impact of data assimilation is not possible without the provision of high quality *in situ* and (high resolution) satellite observations. In order to improve marine forecasts in the decades ahead, these systems must be sustained and expanded with the inclusion of new technological developments. The eddy tracking tool and compositing analysis approach presented in this study is an alternative and innovative validation diagnostic for operational and reanalyses products. In addition, eddy characteristics derived from eddy tracking tools have the potential to become a new mesoscale ocean monitoring indicator (von Schuckmann et al., 2018). Improvements in the CMEMS operational models will contribute to advances in characterisation and understanding of mesoscale physical processes and their role in the functioning of marine ecosystems.

**Acknowledgments**

This work was carried out as part of the Copernicus Marine Environment Monitoring Service (CMEMS) *MedSUB* project (CALL 21-SE-CALL1). CMEMS is implemented by Mercator Ocean in the framework of a delegation agreement with the European Union. Evan Mason was supported by the *MedSUB* project during the investigative and writing phases of this paper. Evan Mason was also supported, in part, by the NASA Physical Oceanography Program award NNX16AH9G. The Ssalto/Duacs altimeter products are produced and distributed by the Copernicus Marine and Environment Monitoring Service (CMEMS)

(http://www.marine.copernicus.eu). We are grateful to Yann Drillet for his helpful comments concerning the interpretation of the eddy composite results.

*Code and data availability.* The eddy identification and tracking code is available at https://bitbucket.org/emason/py-eddy-tracker/ src/default/. The respective CMEMS model data (*GLO*, *MFS* and *IBI*) can be found at http://marine.copernicus.eu/ services-portfolio/access-to-products/. Eddy tracking and composite data and related codes can be made available from the first author upon request.

## Appendix A:  Supplementary materials

### A1    Nonlinearity component maps

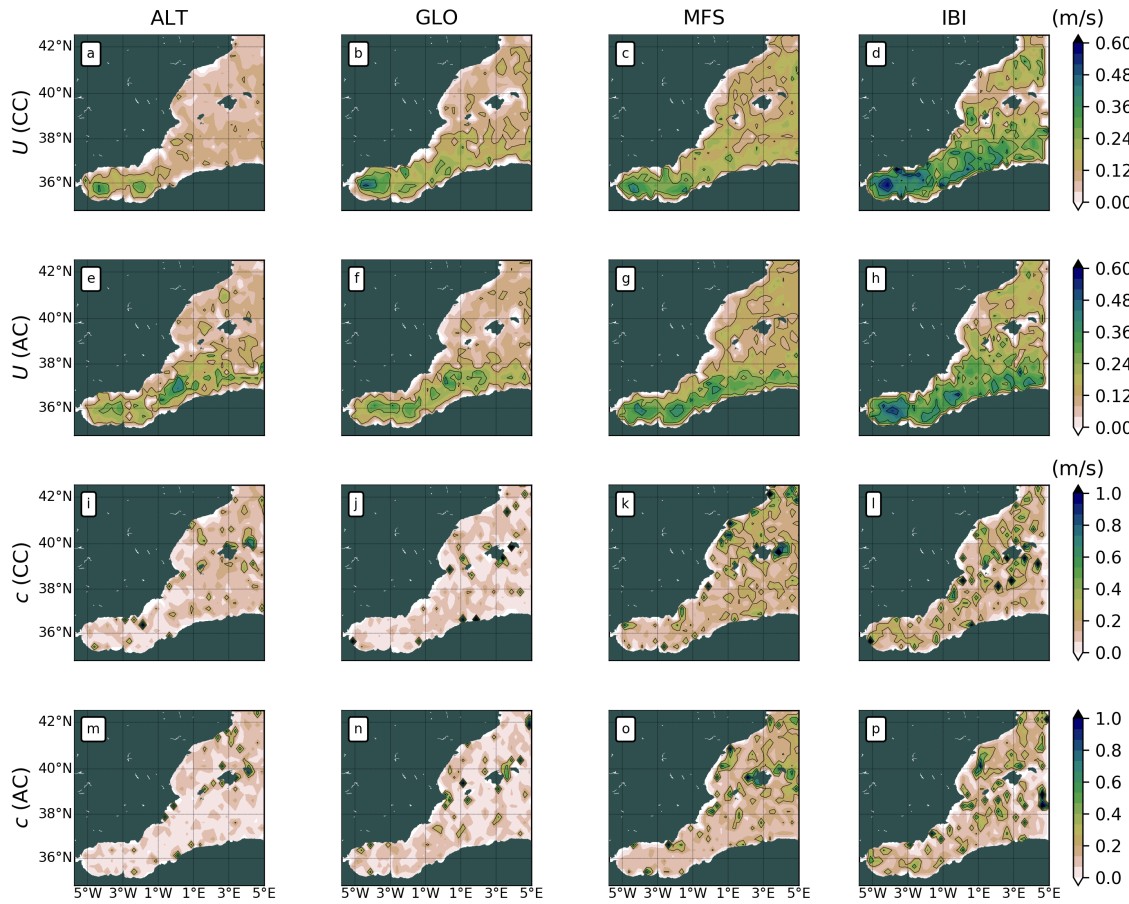

**Figure S1.** Maps of mean eddy swirl speed (a-h) and celerity (i-p) over the western Mediterranean. Columns for *ALT*, *GLO*, *MFS* and *IBI*. Paired rows for cyclones (CC) and anticyclones (AC).

## A2 Tilt correction

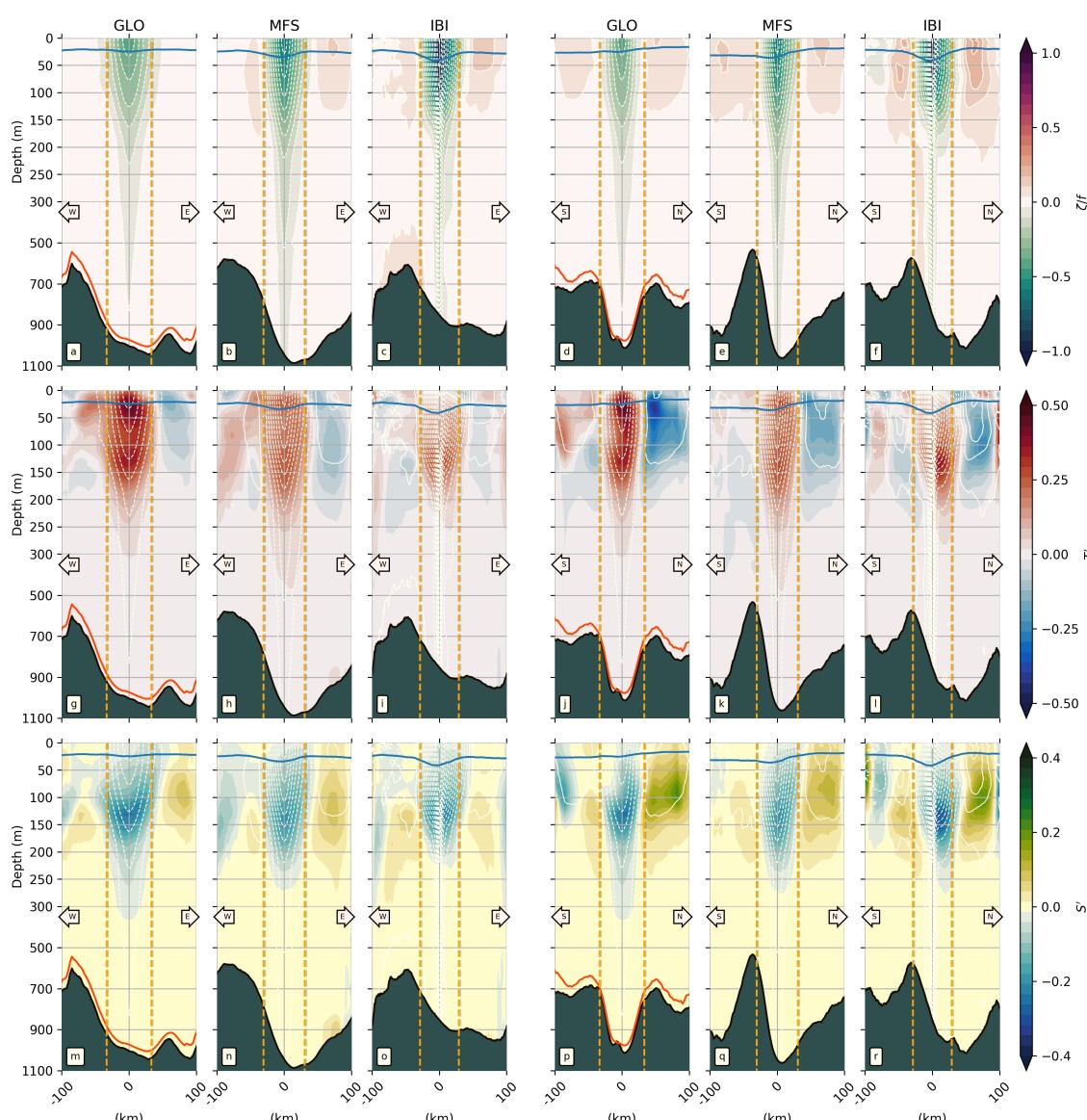

**Figure S2.** Eddy composite sections with the tilt correction described in Sec. 3.6 from *GLO*, *MFS* and *IBI* in the western Alboran gyre. Compare the zonal and meridional sections with those of Fig. 7. Vertical orange dashed lines indicate the boundaries of each composite eddy based on its mean radius estimate from Tab. 2. Composite topographic profiles in black from SRTM, and also in red from *GLO*. Note change of vertical scale at 300 m.

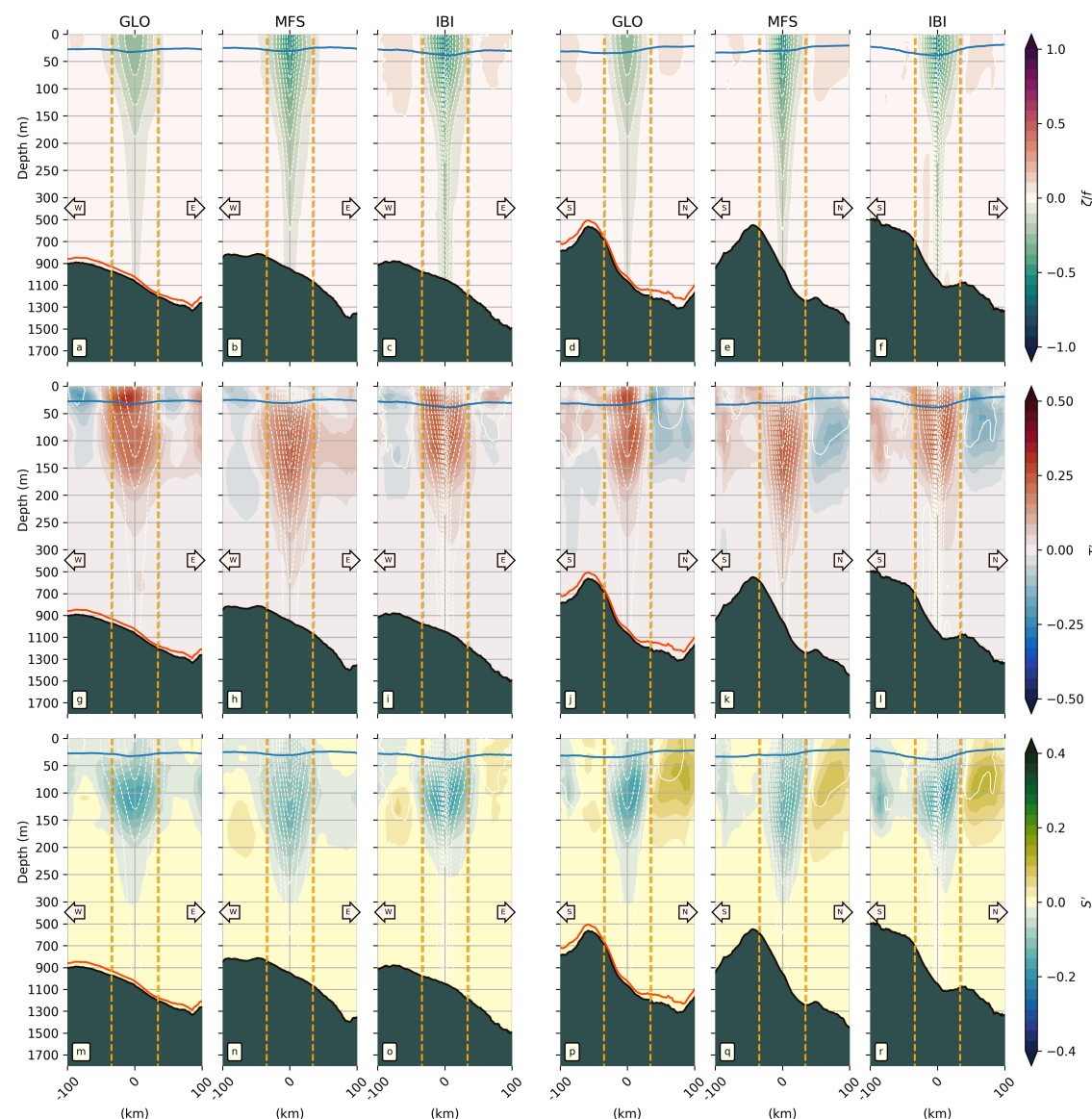

**Figure S3.** Same as Fig. S2 but for the eastern Alboran gyre.

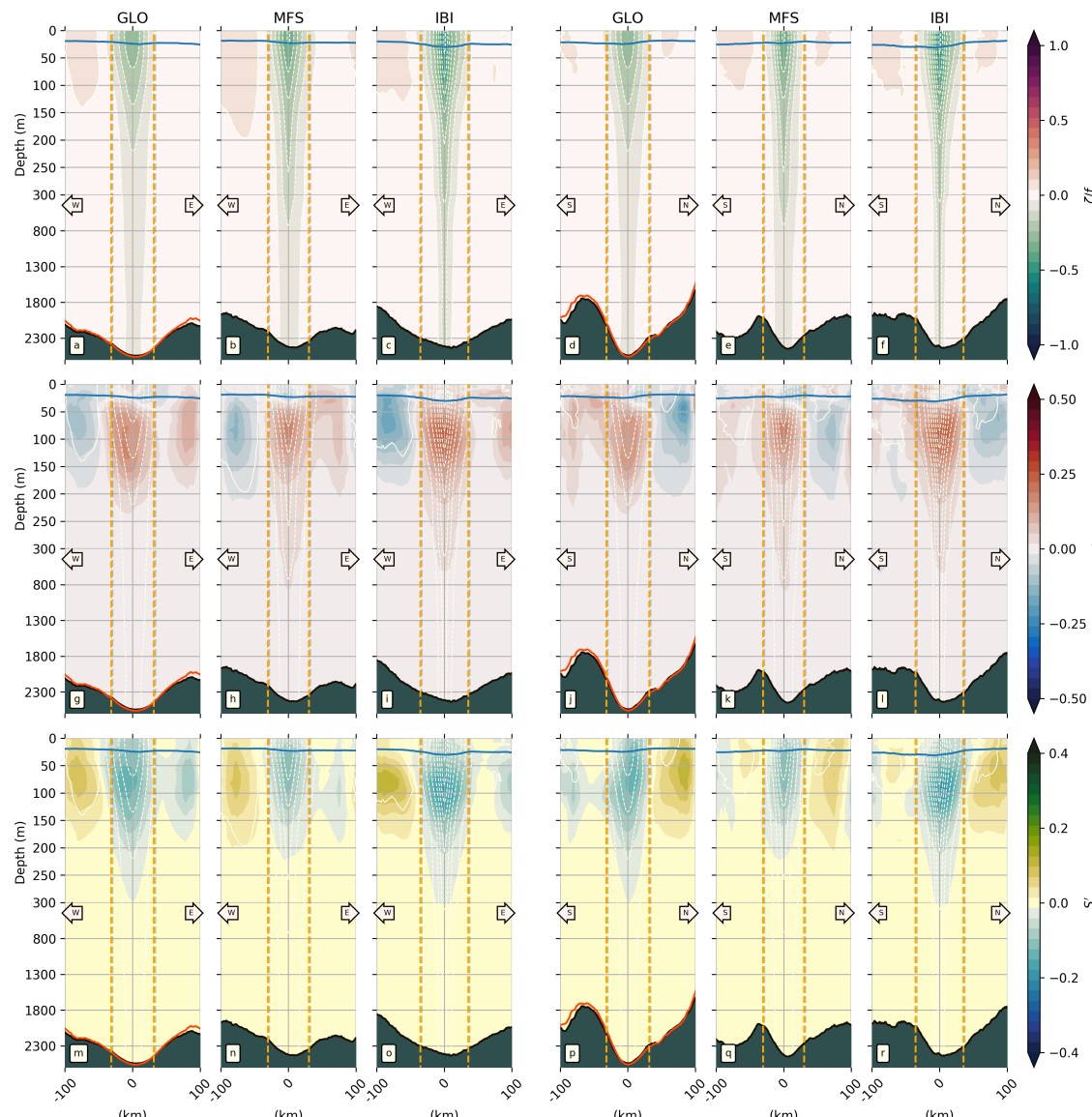

**Figure S4.** Same as Fig. S2 but for the Cartagena frontal region.

# A3 Anticyclonic eddy composite sections

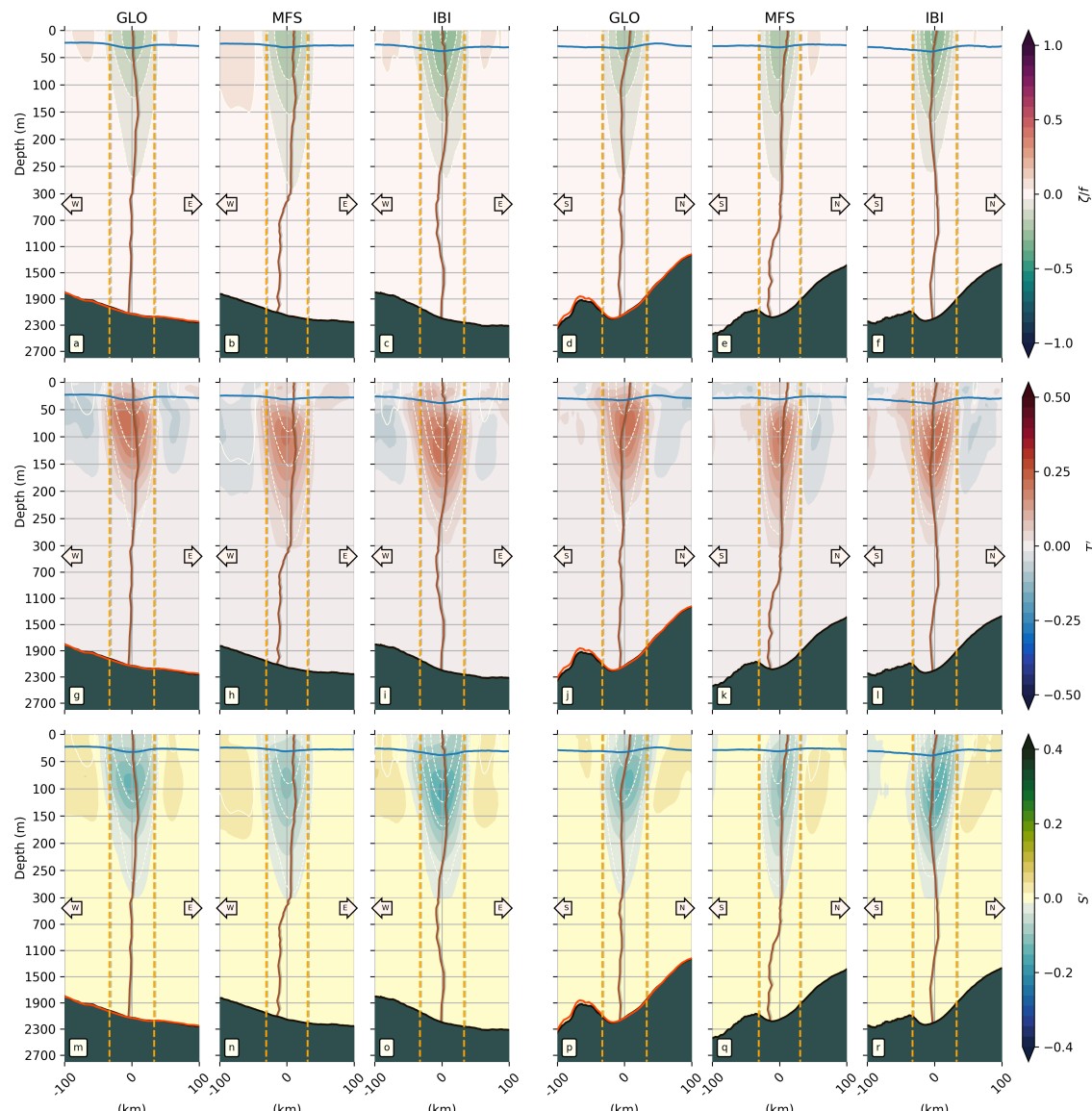

**Figure S5.** Anticyclonic eddy composite sections from *GLO*, *MFS* and *IBI* in the Algerian western region (0°E-
2.25°E, 35.7°N-38.75°N). Left-hand-side (right-hand-side) columns show zonal (meridional) sections of (top to bottom) relative vorticity, temperature anomalies and salinity anomalies, from the surface to the ocean floor. The central position of each section is the median of the longitudes and latitudes associated with the eddy observations used to make the composites. Blue lines indicate the mixed layer depth; the dotted blue line corresponds to the MLD from the *MFS* model. The vertical brown line in each section is the vorticity-based tilt correction (Sec. 3.6). Vertical orange dashed lines indicate the boundaries of each composite eddy based on its mean radius estimate. Composite topographic profiles in black from SRTM, and also in red from *GLO*. Note change of vertical scale at 300 m.

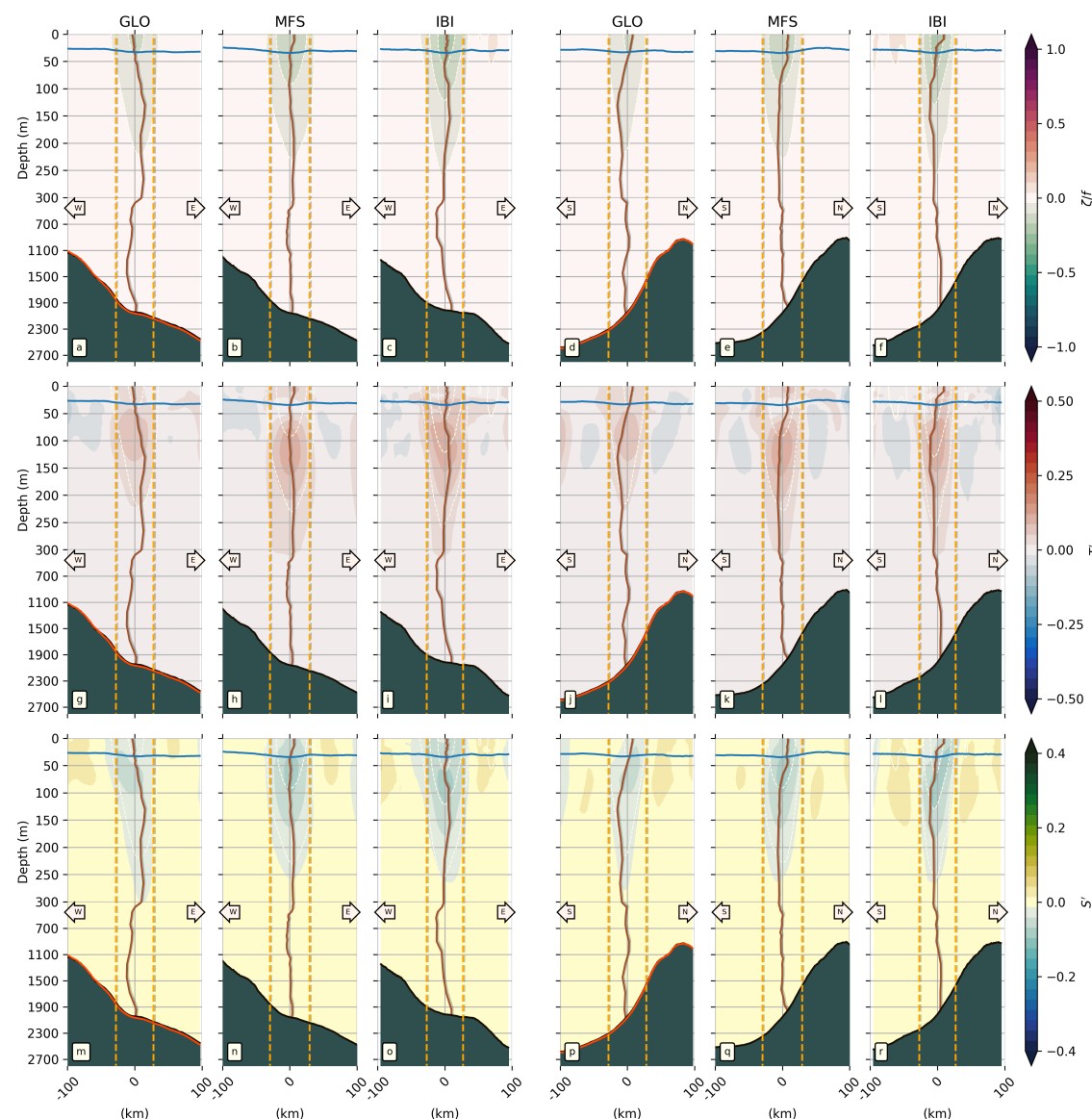

**Figure S6.** Same as Fig. S5 but for the Algerian east (northern) region (2.25°E-4.5°E, 38.25°N-36.65°N).

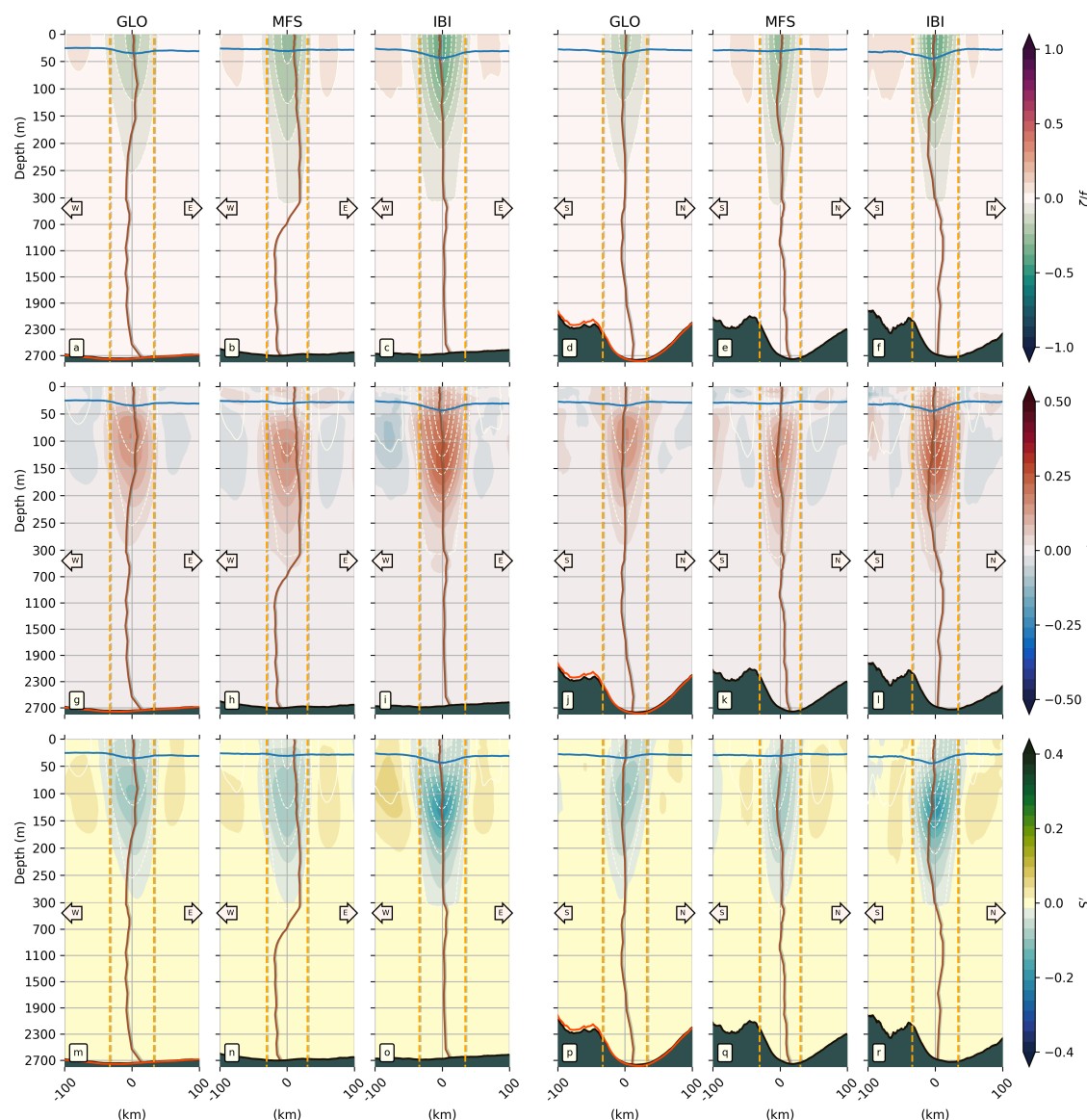

**Figure S7.** Same as Fig. S5 but for the Algerian east (southern) region (2.25°E-4.5°E, 35.7°N-38.25°N).

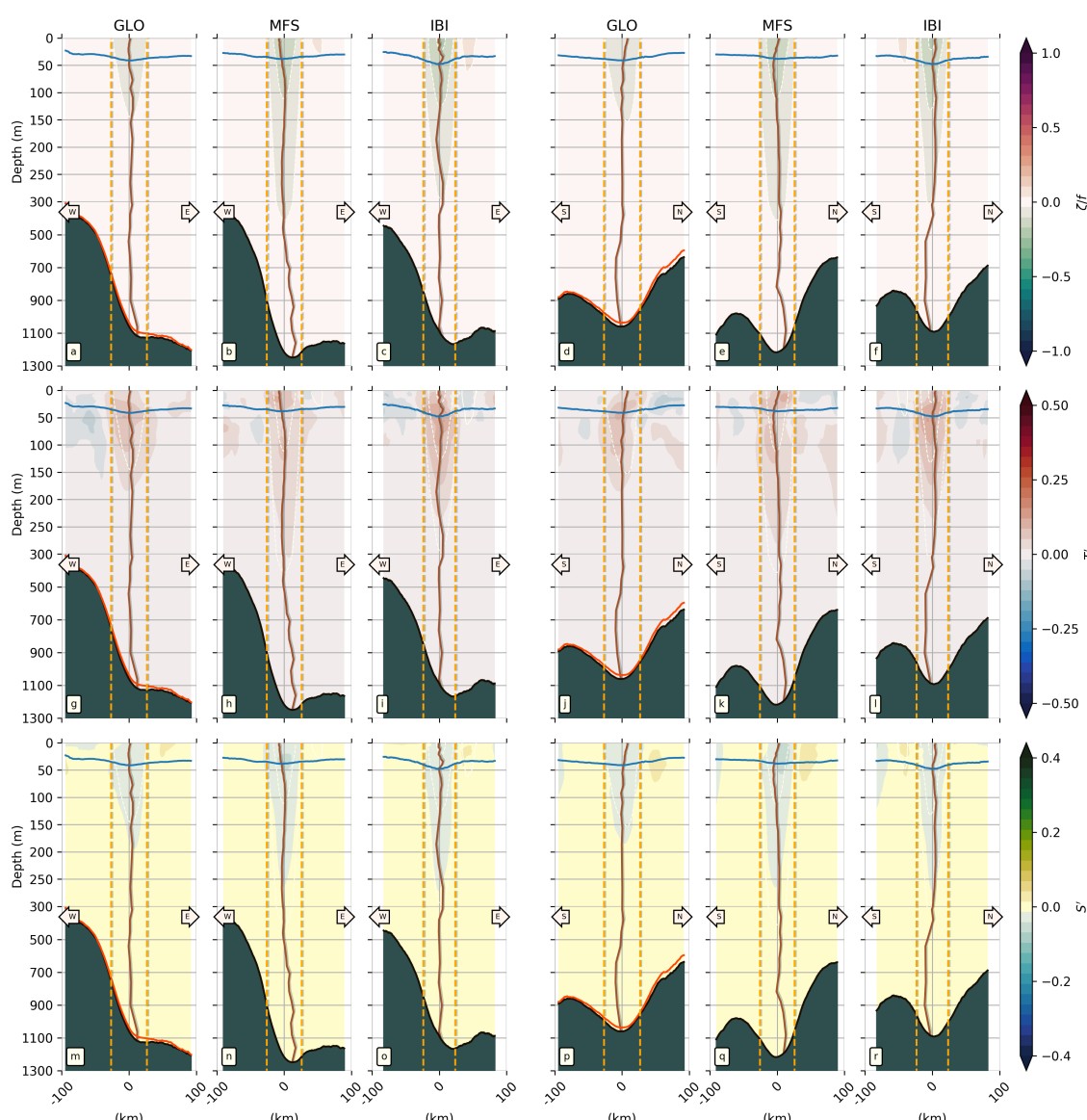

**Figure S8.** Same as Fig. S5 but for the Balearic western region (0.5°W-2.25°E, 38.75°N-41.5°N).

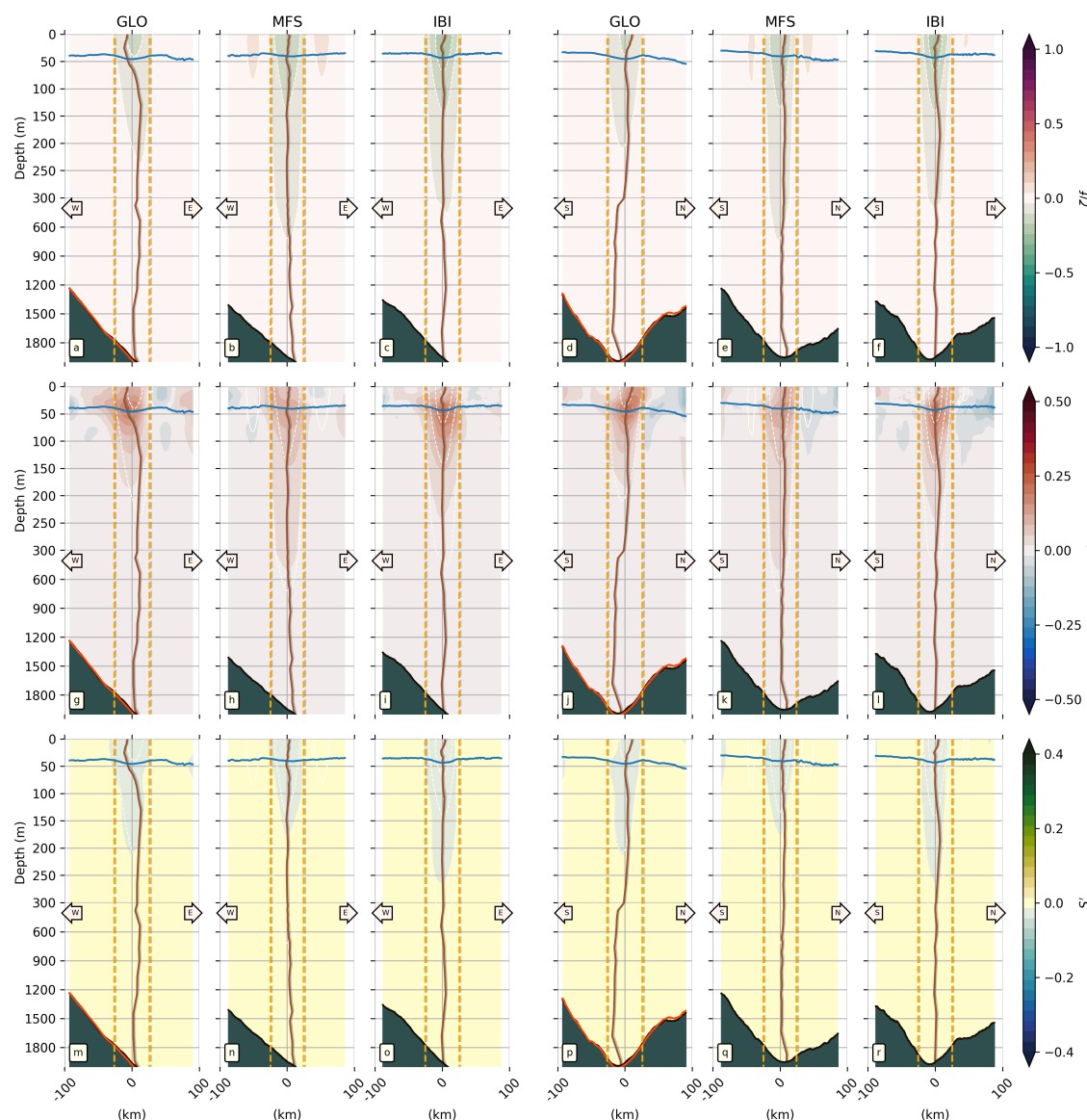

**Figure S9.** Same as Fig. S5 but for the Balearic eastern region (2.25°E-4.5°E, 39.65°N-41.5°N).

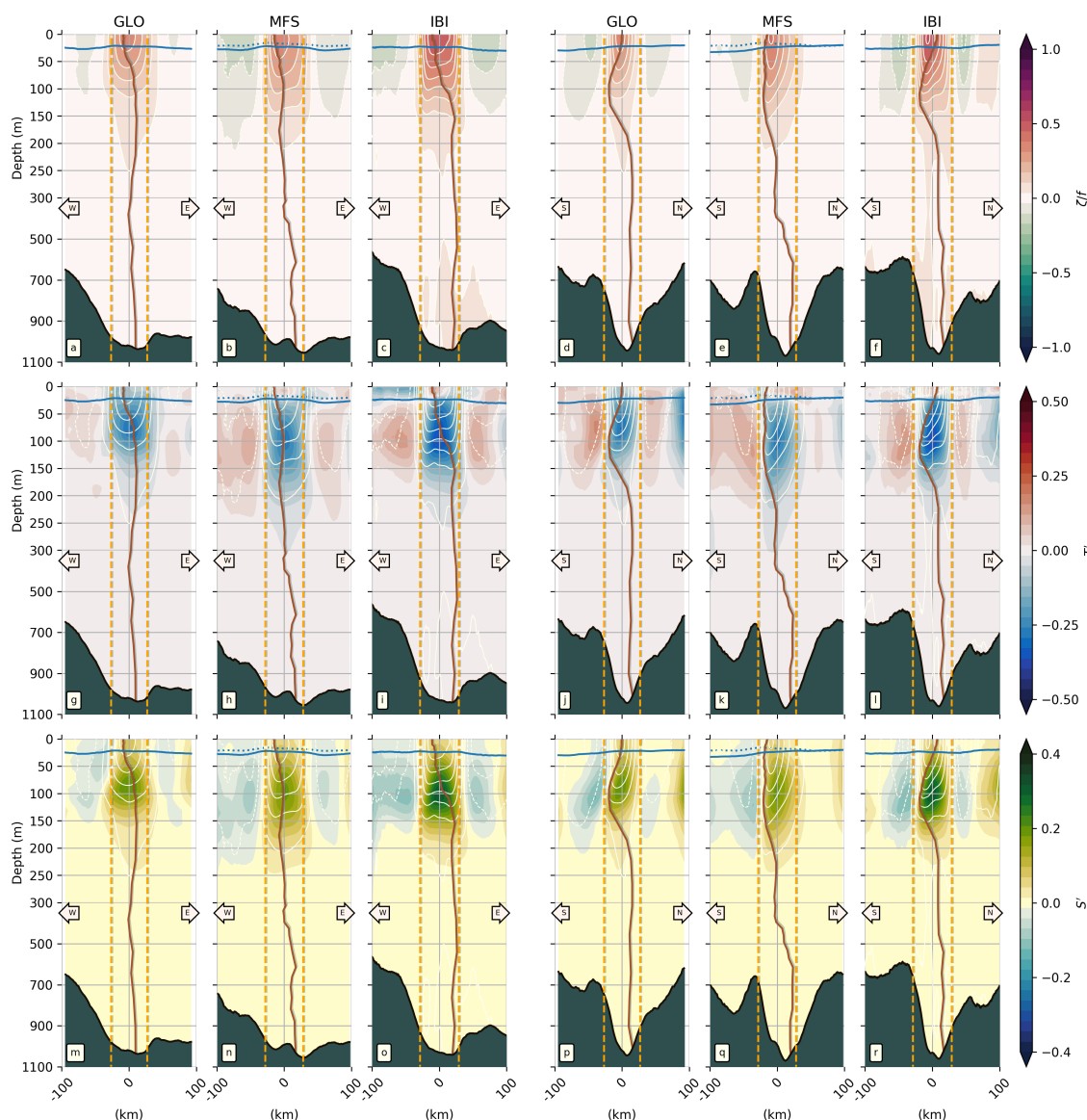

**Figure S10.** Cyclonic eddy composite sections from *GLO, MFS* and *IBI* in the western Alboran gyre (see Fig. 1 for location). Left-hand-side (right-hand-side) columns show zonal (meridional) sections of (top to bottom) relative vorticity, temperature anomalies and salinity anomalies, from the surface to the ocean floor. The central position of each section is the median of the longitudes and latitudes associated with the eddy observations used to make the composites. Blue lines indicate the mixed layer depth; the dotted blue line corresponds to the MLD from the *MFS* model. The vertical brown line in each section is the vorticity-based tilt correction (Sec. 3.6). Vertical orange dashed lines indicate the boundaries of each composite eddy based on its mean radius estimate. Composite topographic profiles in black from SRTM. Note change of vertical scale at 300 m.

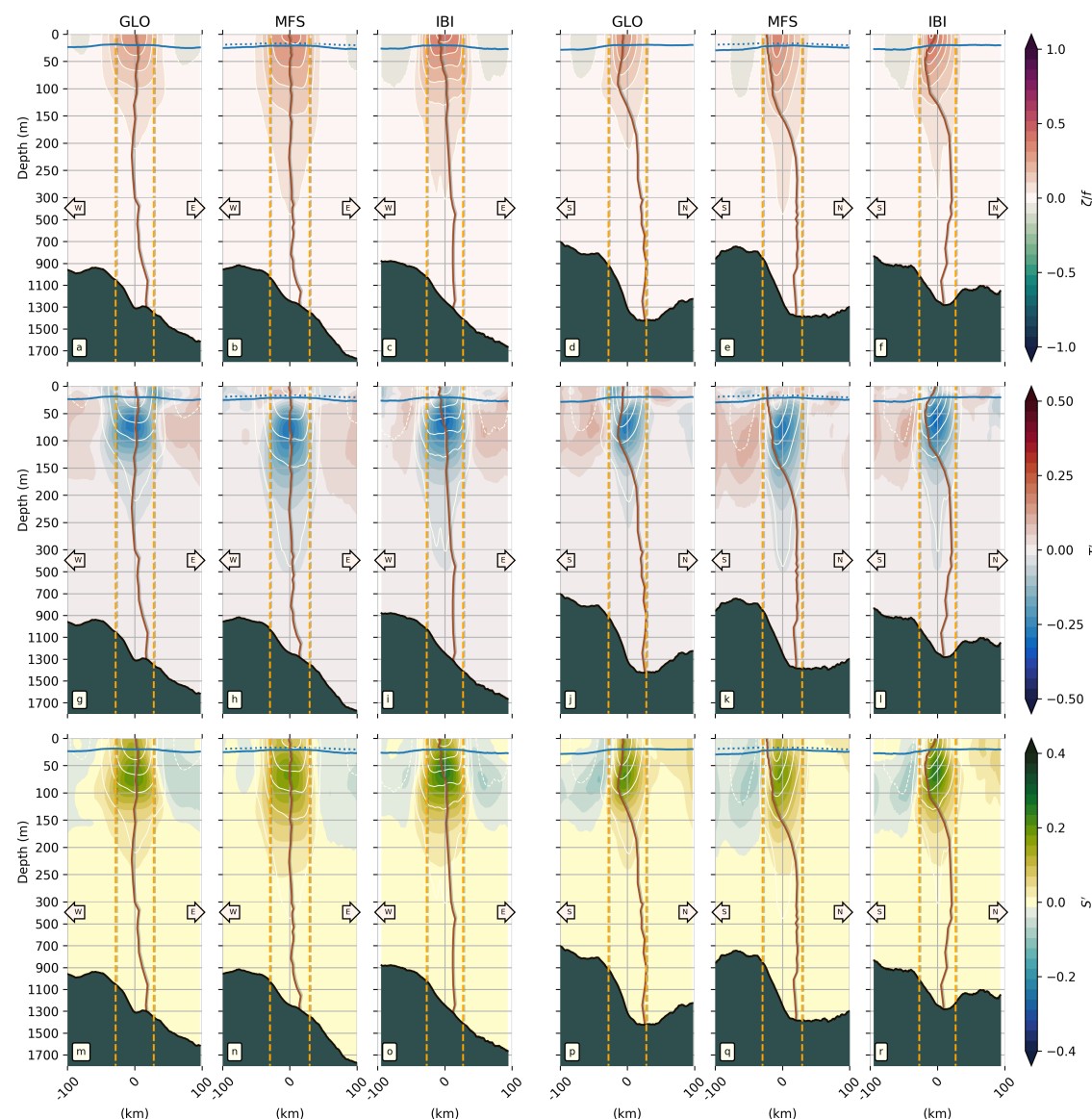

**Figure S11.** Same as Fig. S10 but for the eastern Alboran gyre (see Fig. 1 for location).

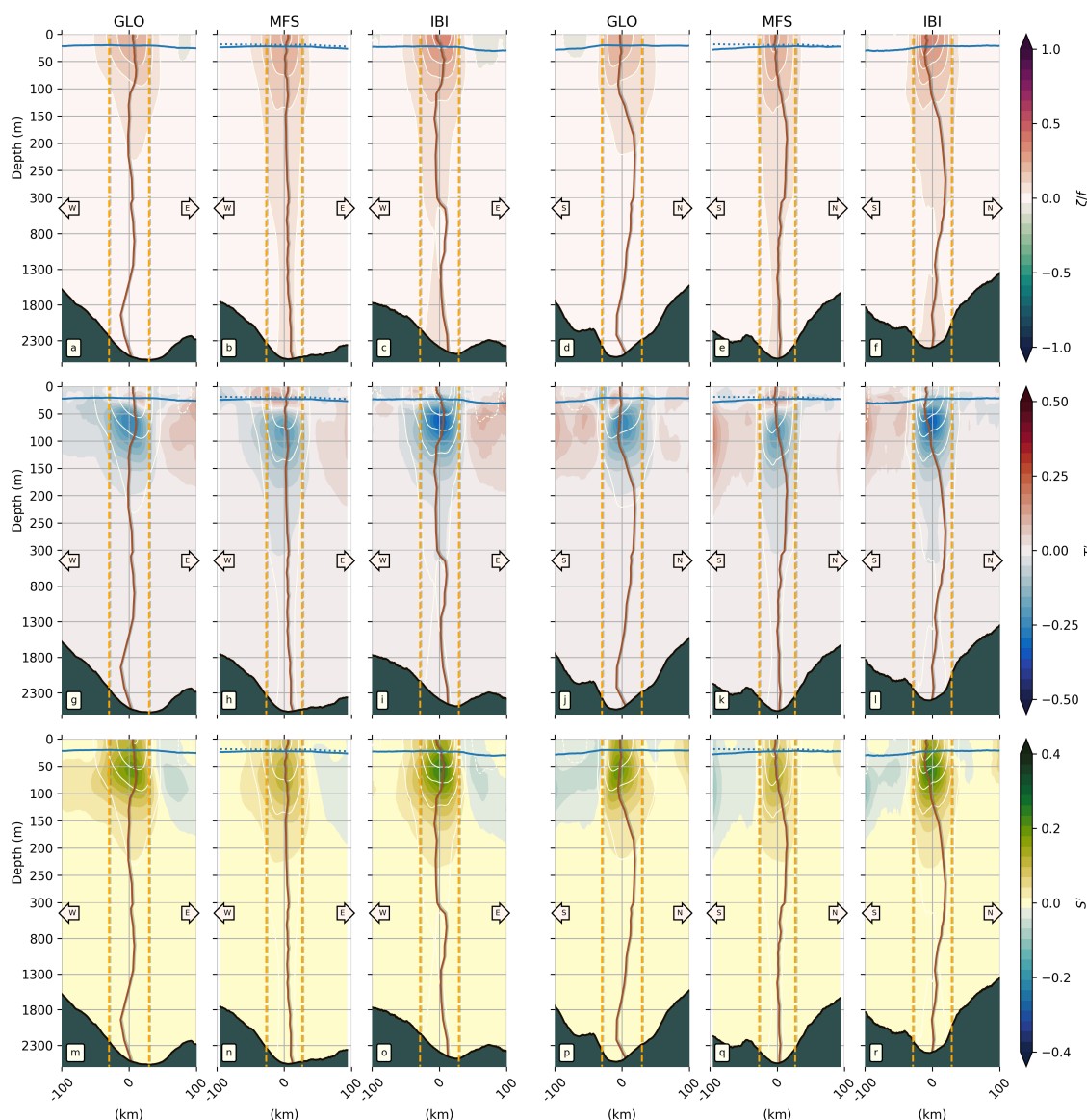

**Figure S12.** Same as Fig. S10 but for the Cartagena frontal region (see Fig. 1 for location).

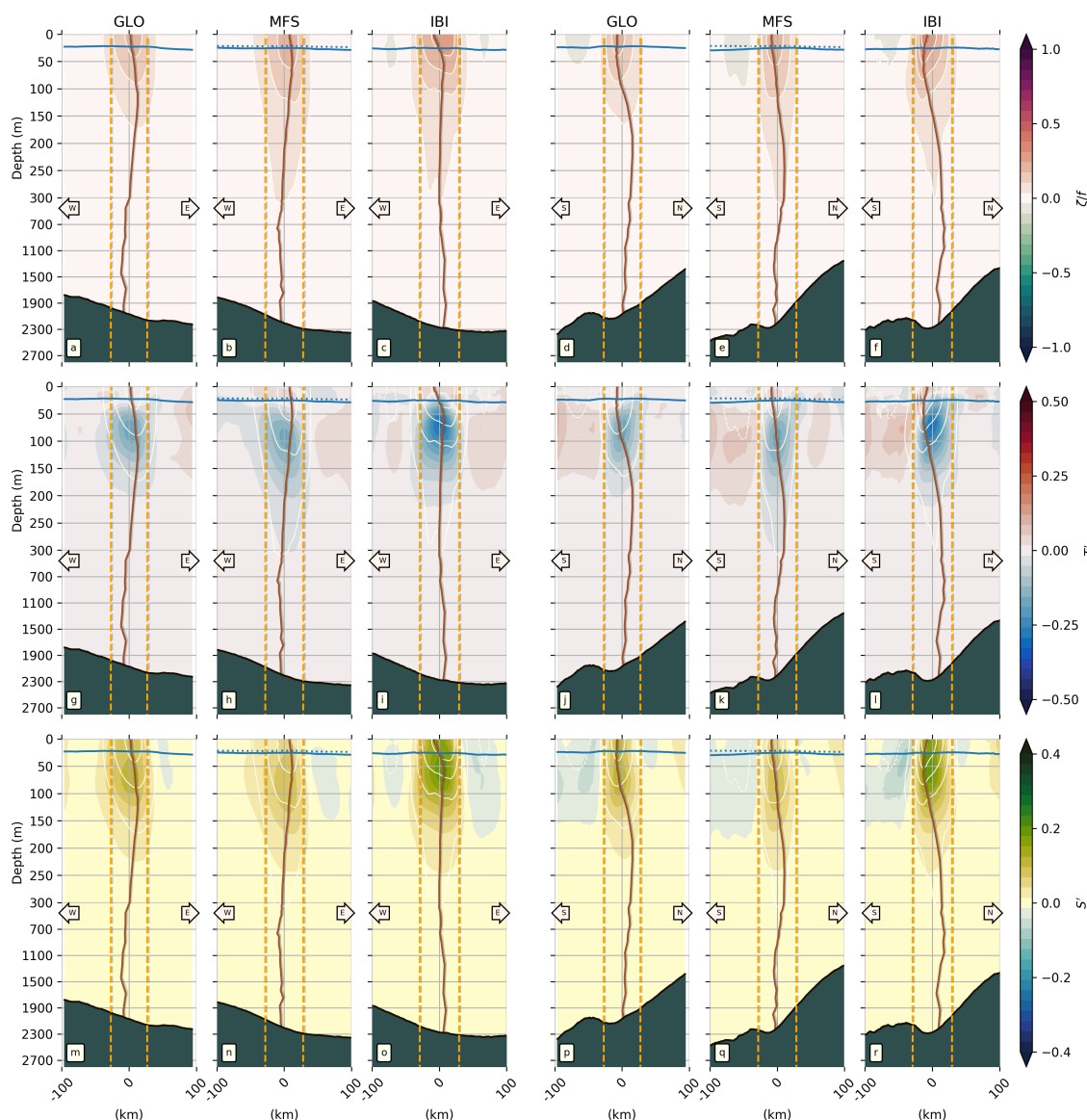

**Figure S13.** Same as Fig. S10 but for the Algerian western region (0°E-2.25°E, 35.7°N-38.75°N).

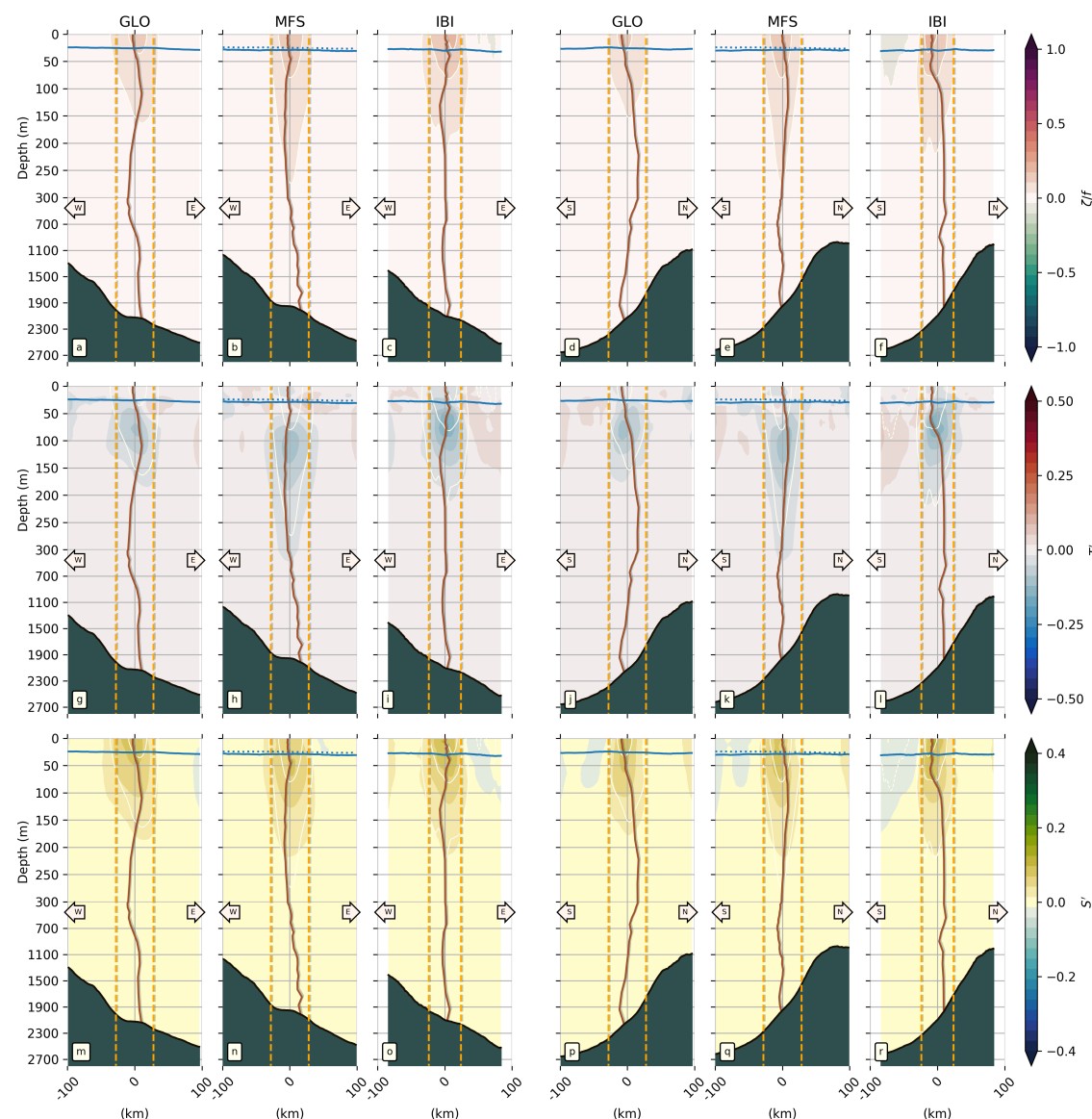

**Figure S14.** Same as Fig. S10 but for the Algerian east (northern) region (2.25°E-4.5°E, 38.25°N-36.65°N).

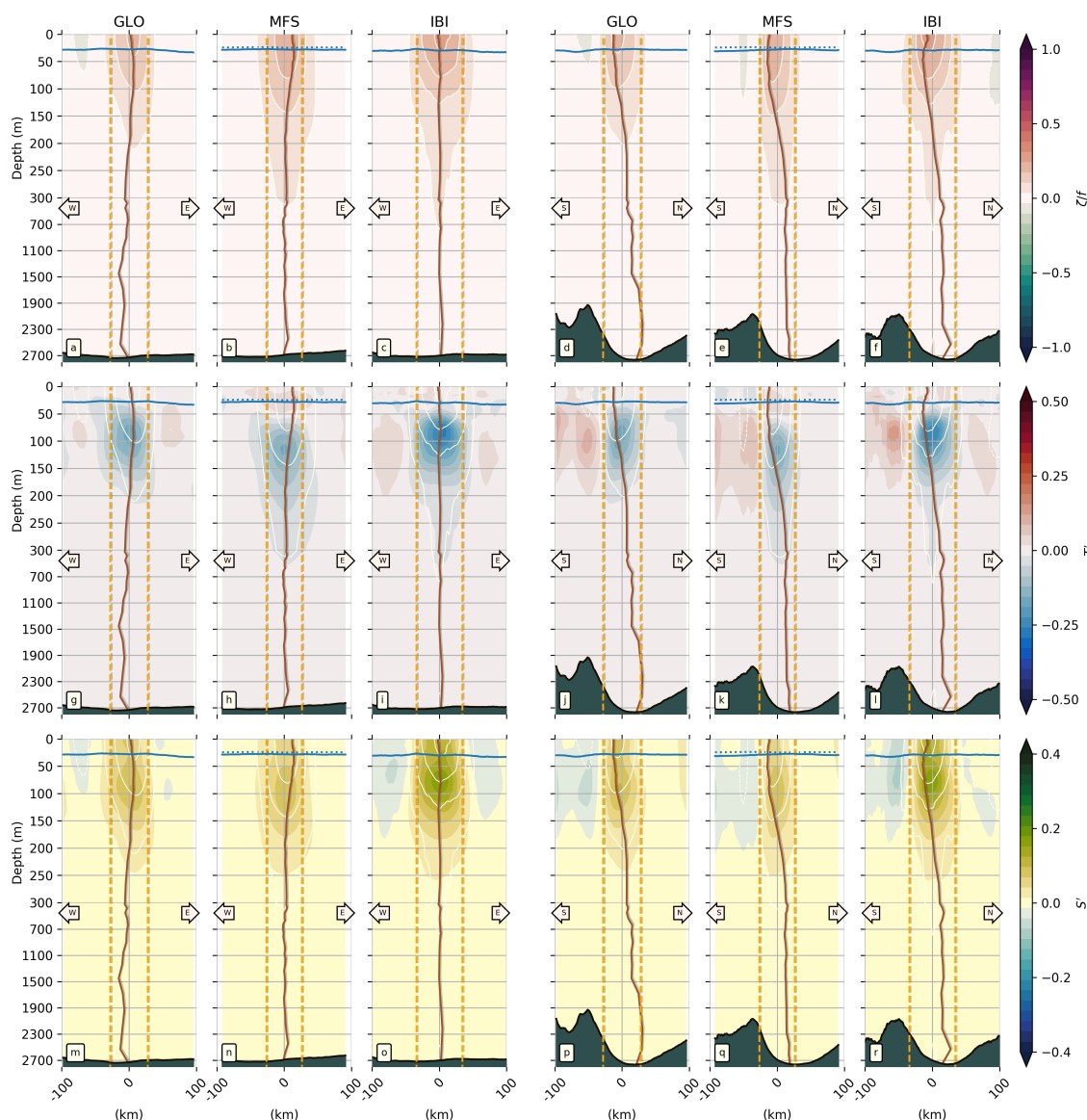

**Figure S15.** Same as Fig. S10 but for the Algerian east (southern) region (2.25°E-4.5°E, 35.7°N-38.25°N).

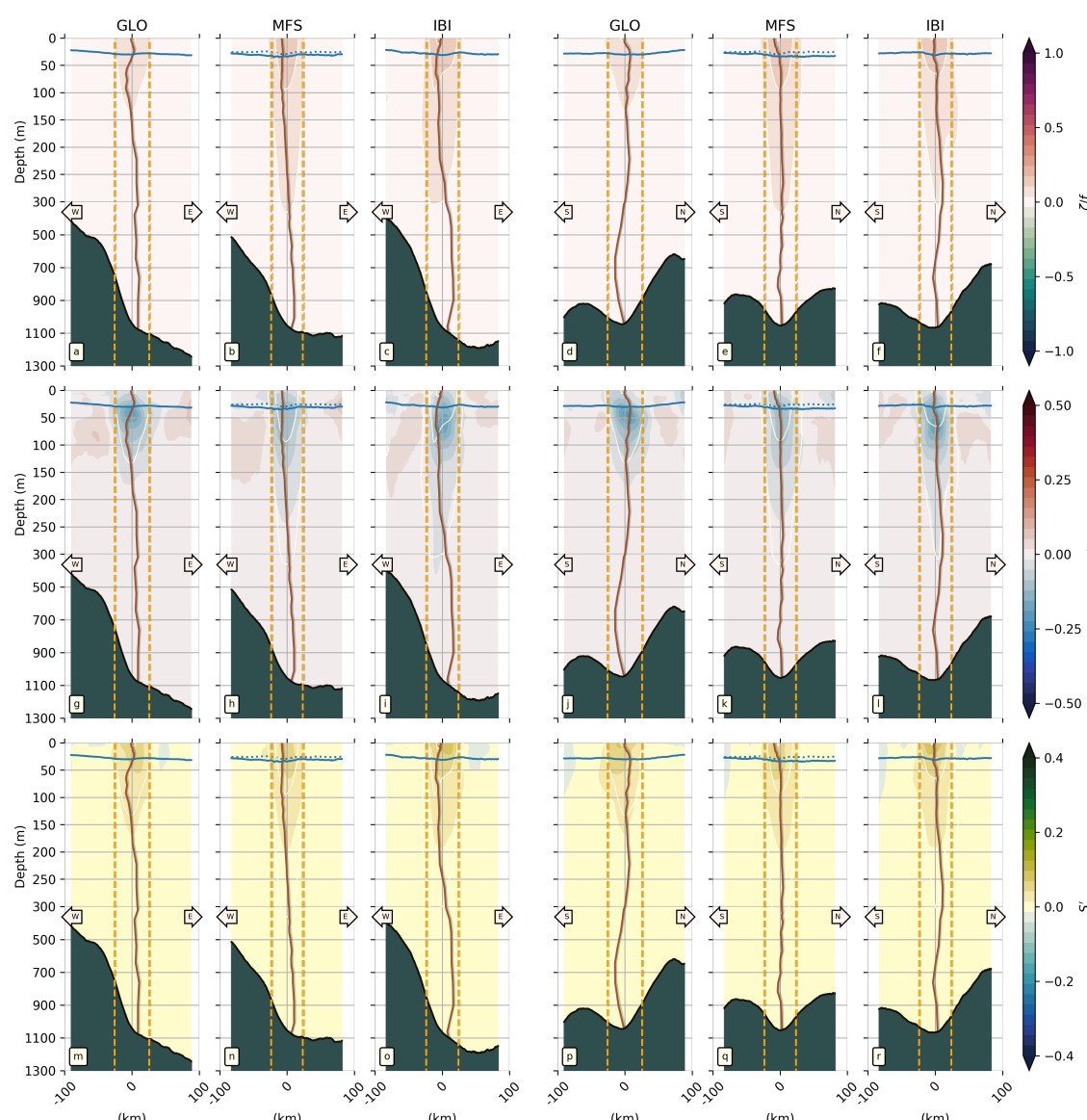

**Figure S16.** Same as Fig. S10 but for the Balearic western region (0.5°W-2.25°E, 38.75°N-41.5°N).

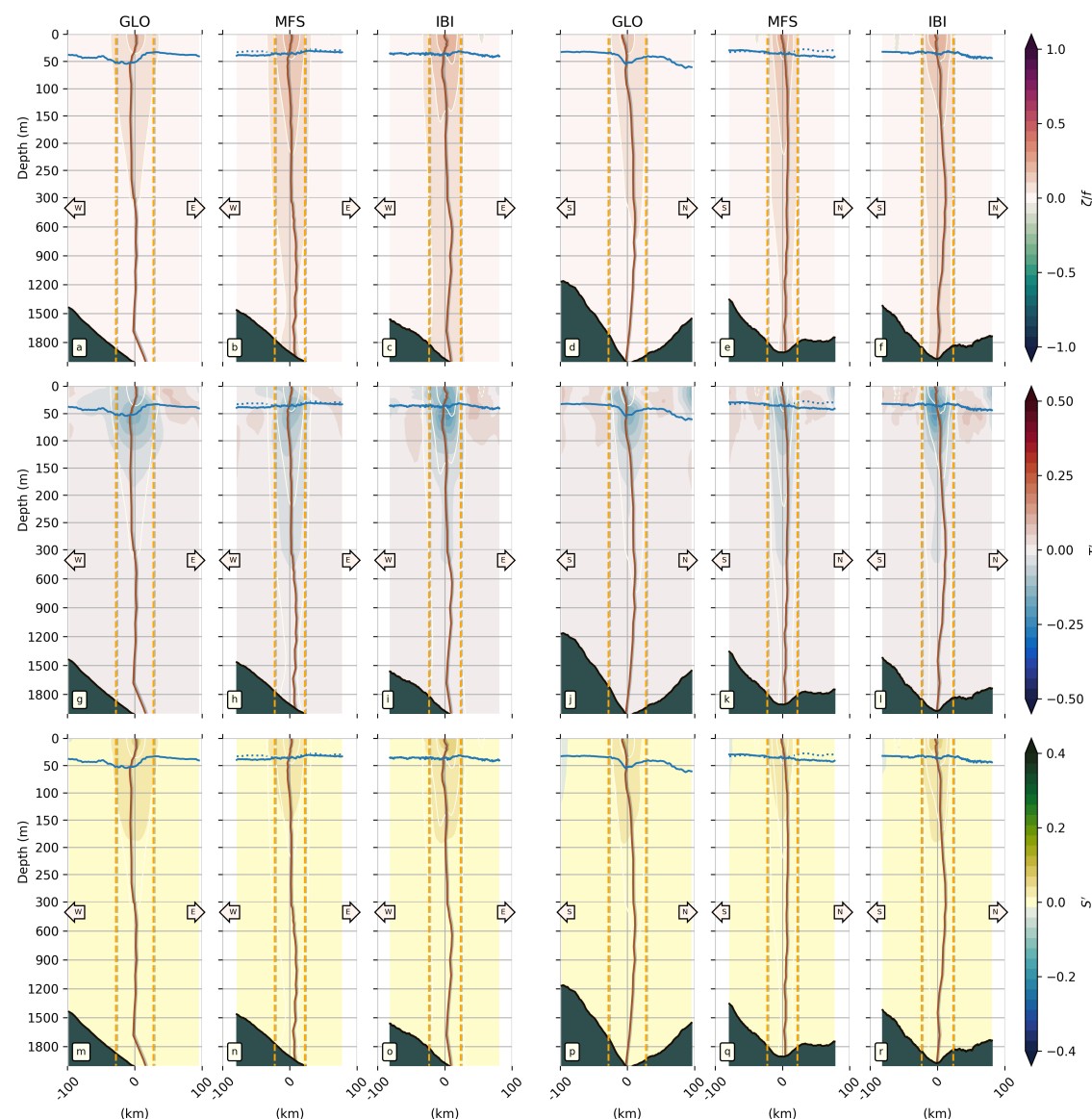

**Figure S17.** Same as Fig. S10 but for the Balearic eastern region (2.25°E-4.5°E, 39.65°N-41.5°N).

## A5 Seasonal mean MLD for the EAG and CRT subregions

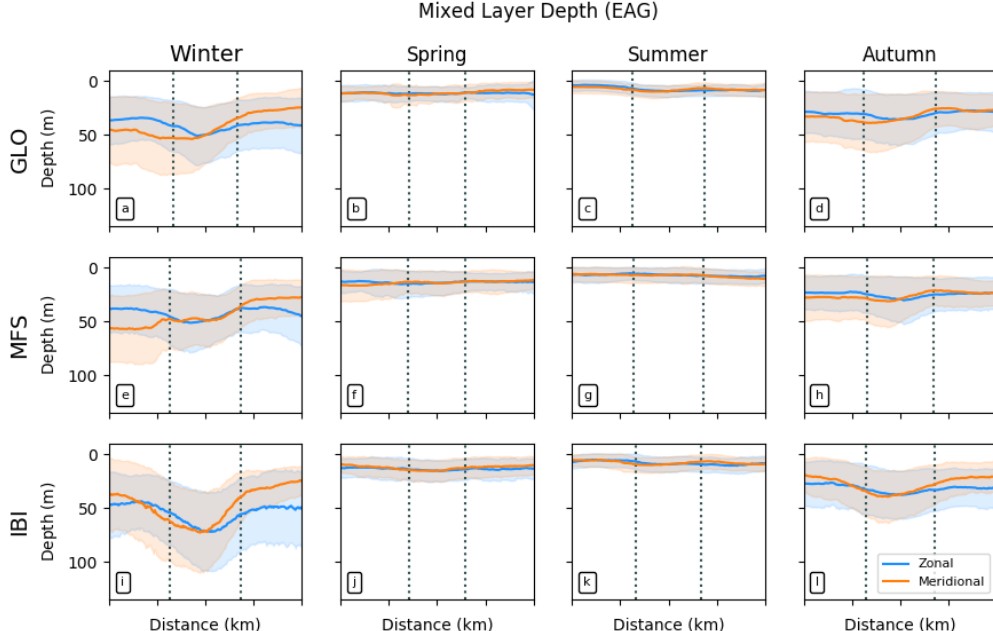

**Figure S18.** Mean and standard deviation of the seasonal mixed layer depth in anticyclones in the eastern Alboran gyre for *GLO* (left column), *MFS* (middle column) and *IBI* (right column). Zonal (meridional) profiles in blue (orange). Vertical dotted lines mark the mean eddy radius from the center.

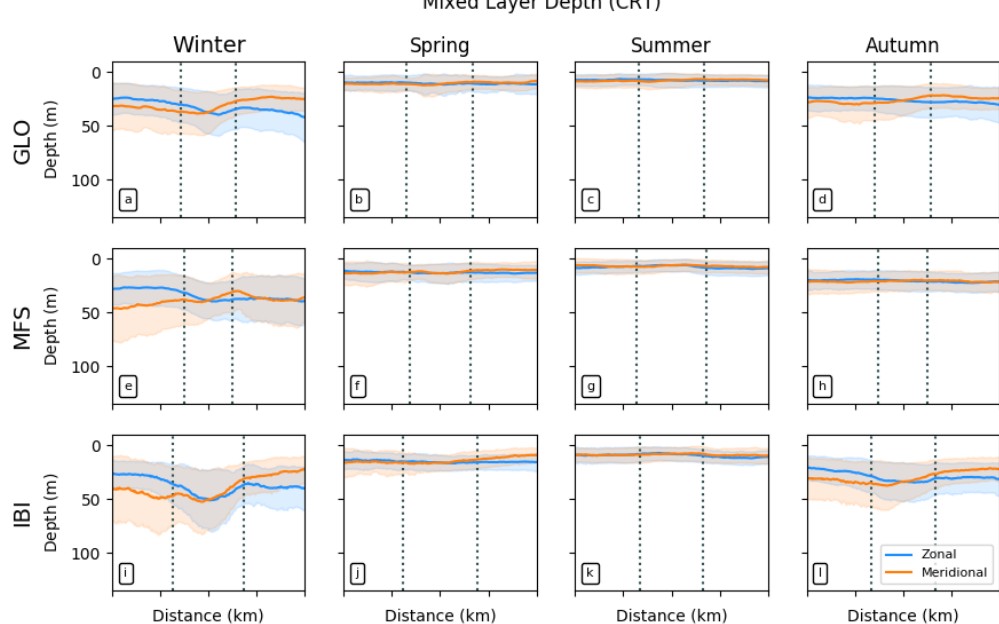

**Figure S19.** Same as Fig. S18 but for the Cartagena frontal region.

**A6    Validation with Argo data**

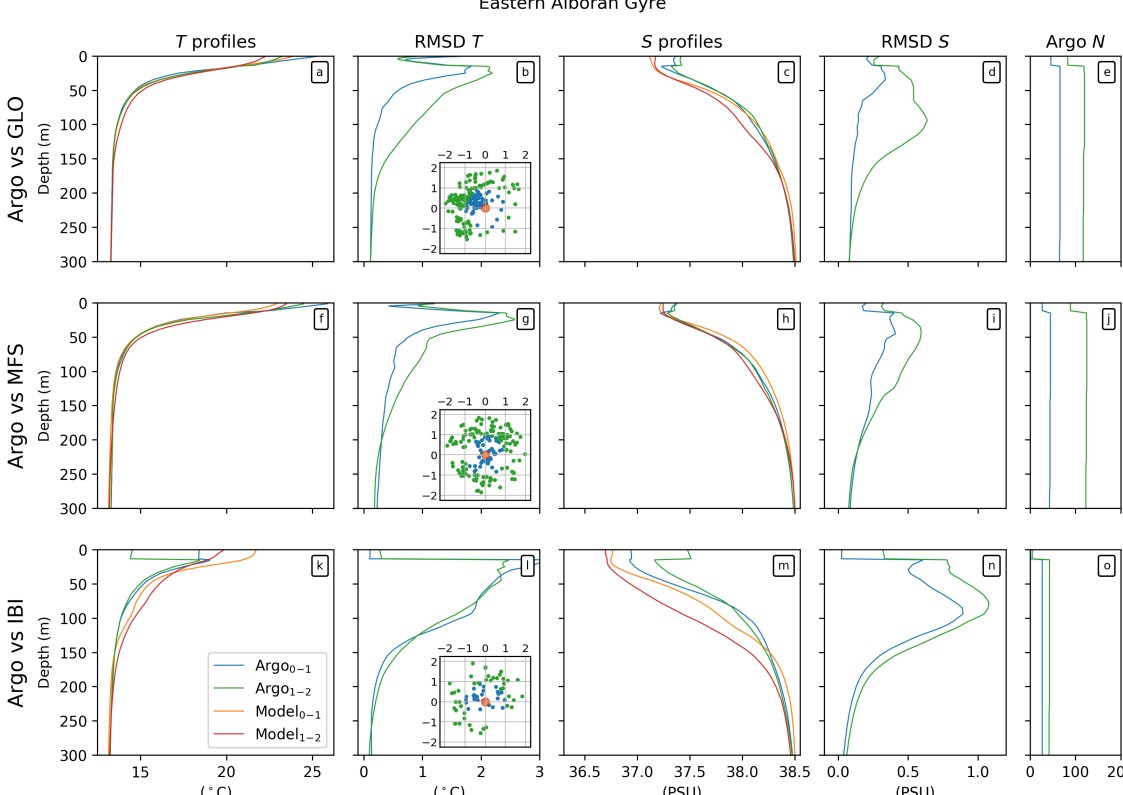

**Figure S20.** Comparison of temperature and salinity depth profiles at the center of each model anticyclonic eddy and its nearest Argo observation in the eastern Alboran gyre subregion. The first column shows mean $T$ and the second the root mean square differences at each vertical level between the individual model and Argo $T$ observations. Argo profiles coloured blue (green) are located inside (outside) the eddy radius, viz. 0-1$L$, 1$L$-2$L$, as indicated in the inset eddy-centric-coordinate maps in column two that show the relative (to the eddy center) positions of each Argo profile. The respective model profiles corresponding to the Argo selections are coloured orange and red. Profiles in columns three and four show the results for $S$. The Argo sample size $N$ at each depth for each model is found in the final column. Rows one to three show *GLO* (top), *MFS* (middle) and *IBI* (bottom).

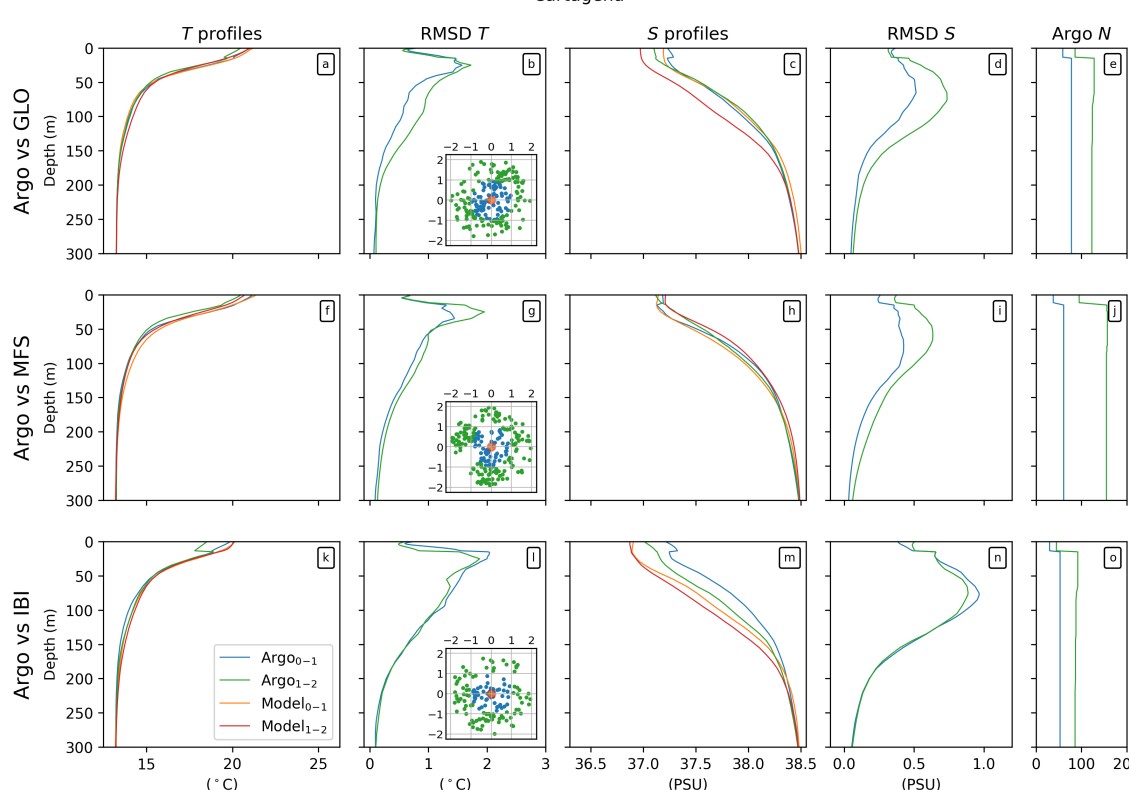

**Figure S21.** Same as Fig. S20 but for the Cartagena frontal region.

*Competing interests.* No competing interests.

**References**

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
