# Peer review of "New insight into 3D mesoscale eddy properties from CMEMS operational models in the western Mediterranean"

_Ocean Science, 2018_

## Referee Comment (RC1) · Anonymous Referee #1 · 30 Mar 2019

General Comments:

In this study, the authors present an eddy tracking tool to investigate the eddy properties derived from three different MFC reanalysis products. A gridded altimetry dataset is used as reference. The main argument for the added value of this study, is that such a tool is useful for model intercomparison, with the objective to evaluate model performance.

The manuscript shows some interesting results regarding the eddy properties for the three different MFC reanalysis products, focusing on the western Mediterranean between Gibraltar and Sardinia, an area governed by intense eddy activity. In my view,

this model inercomparison is valid with such a tool and quite interesting. However, without references to make a proper comparison it is impossible to evaluate model performance between the MFCs, let alone to propose remedies for their implementation. This is a major constrain in the way the motivation of this study is posed. I would advise the authors to reshape their motivation, but not necessarily limit their analysis only in the model intercomparison. For instance, I would like to see a more comprehensive analysis for the differences between products, based on attribution to physics and on the knowledge of the region's dynamics, in order to bypass the lack of a reference dataset. They only thing that the authors should be careful is to clearly stress that they make fair assumptions on how they foresee the future improvements for those MFCs, based on the findings of the eddy tracker. After all, some of the co-authors are leading the developments in those MFCs and have a good knowledge of the systems they operate. In the same line of thinking, the draft is too descriptive focusing on the eddy properties and the model intercomparison, without the authors attempting to discuss in details differences in physics between the MFCs products.

Finally, one additional problem with the manuscript is that it is sloppily written, as if it was copied/pasted in a rush manner from another draft or report. The figures are not properly introduced in the text and I had to guess throughout the whole review in which figure the authors are referring to. Overall, I find the manuscript worthy of publication in Ocean Science, after a major revision. Please find below a list of comments following the flow of the text, that the authors should address in the revised manuscript.

Specific comments:

1) In this study, an eddy-tracker modelling tool is used to evaluate other models. How confident the authors are that their approach is valid and what are the limits of the method? In the Introduction section I felt that there are not enough references to link this approach with previous studies.

2) In the Abstract, the authors stress the fact that "This information can be informative

for the ongoing development". I agree and I was eager to see such an analysis in the Discussion and Conclusion section, but in my view the authors failed to do so. I think that such an analysis is important and the authors should provide a more comprehensive and detailed discussion.

3) Page 2, line 16: What is "STA (2017)"?

4) Page 3, line 7 and elsewhere in the text for all the figures: "Fig. 1.1". This is not the proper format to mention the figures and should be changed everywhere in the text. In addition, I 've noticed that some figures should have been mentioned earlier in the text with respect to other figures, or other figures are not mentioned at all (e.g. Fig. 2 seems not to be mentioned in the text, unless it is Fig. 2.1, but then should be mentioned earlier). In addition, all figures in the supplement material are referred with question marks, and it was impossible to understand which is which to the point that it become annoying.

5) Page 3, line 21: "that extend a horizontal distance of 8x the eddy radius". Why? It seems to me that an explanation or a reference is missing here.

6) Page 5, lines 12-13: "Delivered products are bilinearly interpolated onto a regular lon/lat 1/36 grid". This is a major issue for the analysis that follows. Why to interpolate coarser MFS and GLO products into the higher resolution IBI and not leave it as is or make a coarsening of the IBI and MFS to GLO resolution? I guess this is also linked to the tuning of the eddy tracker, but an explanation is missing here and this fact compromises the findings of this work. The authors should elaborate us regarding this pre-processing interpolation. Perhaps, as a supplement material the authors can provide one example for the opposite technique to coarsen IBI and MFS into 1/12 GLO resolution and perform the eddy tracker (with same/different tuning?).

7) Page 5, lines 13-17: "The stated general...ocean dynamics". This is a very general statement that is redundant in this section. Remove or include it in the Introduction.

8) Table 1 IBI ETOPO definition is presented with question marks and is missing.

9) Page 8, lines 1-3: "The same... will differ". How valid is the fact that the authors use the same tuning for different resolution products (this is also related to the pre-processing interpolation at 1/36; see my previous comment 6).

10) Page 8 in section 2.3: the definitions Le, A and EI should all be declared one line earlier I think.

11) Page 8 section 2.4: Is the composite grid at 1/36? Calculations of dynamical properties (e.g. vorticity) are first calculated on the native MFC regular grids as delivered through the CMEMS portal and then on 1/36? Clarify in the text.

12) Page 10, line 5: "...and their range variability is shown in Sec. 2.6". We are in Sec. 2.6, what I have missed here?

13) Page 13, line 25: "...property anomalies of the...". Anomalies with respect to what? Clarify in the text. The term "anomalies" is also used elsewhere in the text and a clarification is needed there as well.

14) Pages 15-16 "The model increase can be explained by the increasing model resolution". Bad phrasing. Rephrase in the text.

15) Page 16, lines 2-3: "in IBI is suspected...may be an overestimate". Why do you suspect that, is there any reference or an explanation?

16) Page 16, line 26: "...at each vertical level". Is there some sort of vertical re-gridding in the post-processing for the different products pertaining to the fact that the MFCs have different vertical grids? Clarify in the text.

17) Page 18, line 2 (and Table 2): "...are around 9% smaller than the means...". The radii median is always found smaller than the radii mean. Does this means something for the eddies properties, shape and distribution?

18) Page 22, lines 19-20: "As shortwave solar radiation increases from spring to summer (Ruiz et al., 2008),...". Do we really need a reference for this statement?

19) Page 24, lines 15-17: "The most likely hypothesis...acts to cool the surface mixed layer". Why not Data Assimilation, how can you tell? How do we know that IBI has the best performance in terms of upper ocean temperature due to tides, when the other MFCs include SST DA? I guess you should rephrase this part of the text and include DA in your fair assumptions. In addition, in the continuation of this paragraph the authors attempt to make some recommendations to MFCs, based on tides and bulk heat fluxes. I am OK with that, but I think there are also other factors that should be mentioned here, such as vertical parametrizations, turbulent closer schemes, vertical stretching and number of levels (especially when the MFS has increased by a lot the number of vertical levels in the new version, hinting perhaps to better performance?) etc. Finally, suggesting that the GLO MFC should have tides is a logical but not straightforward suggestion, because of the complexity in doing so, and because the area under investigation in this study is very small to derive any strong conclusion for the global configuration.

20) Regarding the authors comment for the ARMOR3D product in the Discussion and Conclusion section, I agree that without such products is difficult to assess how close to reality the eddy composites are. However, even if you had this product on its coarse resolution (if not mistaken at 1/4 of a degree?), that would still not be sufficient for your domain of application, so I guess the ARMOR3D product for regional applications should be delivered on a higher resolution grid? Perhaps the authors want to comment on this.

Best regards.

---

## Referee Comment (RC2) · Anonymous Referee #2 · 3 Apr 2019

**General Comments:**

In this manuscript, the authors applied an eddy tracker and an eddy compositing technique, to three forecasts models: (i) the Mediterranean Forecasting System (MFS), (ii) the Global Mercator model (GLO), and (iii) the Iberia-Biscay-Ireland system (IBI), provided by the Copernicus Marine Service (CMEMS) website. They used the py-eddy tracker, an eddy detection and tracking method similar to the procedure proposed by Chelton et al. (2011). This method is widely used by the scientific community. The eddy compositing analysis is applied to analyse the 3-D structures of three gyres of the Alboran Sea (WAG, EAG, CRT), an interesting area of the Western Mediterranean,

characterized by strong density gradients and high mesoscale activity. The authors compared the models results to evaluate their performance and conclude the paper with some technical suggestions in order to improve these CMEMS products. This paper is well structured and the methodological approach is correct for the purpose of the journal. Nevertheless a more detailed description of the methods and additional results are needed to support the interpretation and conclusion. Therefore, at this stage, a major revision is required before acceptance. Please see below for more details:

Major remarks:

1) I found the Section 2.4 (Eddy compositing) hard to follow. The methodology used for the 3-D eddy composites and for the computation of the anomalies of T and S is not adequately explained. The authors refer to the article of Mason et al., 2017 (see pg. 8 line 15) but it is applied in a different context and in a different area.

2) The eddy properties, detected by py-eddy tracker applied to the three models and to the ALT data, are not adequately compared (section 3.1). In Section 3.1.1, the results on the numbers of eddies detected and tracked in relation to their lifetimes are not quantified. In Section 3.1.2, the authors have identified similarities "between the patterns" of eddy radius, amplitude and intensity (pg. 10, I.16, pg. 11, I.1 and I.11). These may be due to the strong signal of the Atlantic Water flowing eastward, which generates the large and energetic Anticyclonic Eddies. This flow causes differences in terms of eddy properties between the northern and the southern part of the basin. Pessini et. al, 2018 identified these N-S differences as function of the area of formation and lifetime. Escudier et al., 2016 compared the eddies distribution per radius and lifetime, of altimetry and model data. I suggest the authors to follow the last approach since, in my opinion, more detailed comparison between the models and ALT data are needed.

3) The authors did not describe how they calculated the mean and median coordinates/radii of the eddies, listed in Table 2. There are various methods to calculate them OSD
for example: (i) selecting all the eddies detected by the eddy tracker independently for lifetime, (ii) discriminating for lifetime (i.e. avoiding the shortest) and (iii) selecting only the eddies identified simultaneously by the three models in the same sub-area. The 3-D eddy composite analysis should depend on these variables (pg.3, I. 10 to I. 22) and therefore should be method-dependent. Some results (pg. 18, I. 11 to I.12) and conclusions (pg. 24, I. 13 to I. 21) are based on vertical structure of T'. The authors actually cannot demonstrate "how closely these eddy composite results may correspond to reality" (pg. 26, I.10-11). For these reasons, a check to verify if the 3-D sub-region eddy composite results are method-dependent should be done. Alternatively, I suggest the third method because is the most appropriate to compare the model results.

4) In Table 2, please provide the mean and median coordinates and the properties of the eddy for the altimetry data (section 3.2). This will allow a comparison among the models and the altimetry data.

5) Table 3 is not acknowledge through the text. The variables Rmin, Rmax, Rmean, Rmedian, Rmad are not defined. Some values are missing and other seem to have no sense. Specifically, in "Product: MFS" at line "CRT" are listed the values: 111, 222,333, 444, 555 which seem to be out of scale. In Section 2 (Data and methods) the effective and the speed-based (inner) radii are indicated respectively with the symbols Le and L while in Table 3 are labelled as R.

6) In Figure 10, the seasons are not mentioned. In winter and autumn, the deepening in the mixed layer depth (MLD) in the center of the anticyclonic eddy increase from GLO to IBI (fig. 10). This seems coherent with the zonal and meridional relative vorticity of figures 7 and 8, having more intense anomaly for higher resolution models. I assumed that these differences in the MLD inside the anticyclonic eddy are due to rotational speeds, better simulated in these higher resolution models. Anyway the authors assert that "the incoming Atlantic Jet in IBI is suspected to be too strong such that these  $\zeta$  values may be an overestimate (pg. 16, I. 2-3)". The last sentence should be motivated and this part must be clarified also because these considerations could be

OSD
used to evaluate the performance of the models (pg. 24, I. 22) and to provide a more consistent conclusion.

Minor remarks:

7) In Section 2.1 (pg. 5, I. 10-11), the authors state that the variables, for the period 1 January 2013 to 30 June 2016, are downloaded by ftp from the Copernicus CMEMS portal from: GLOBAL\_ANALYSIS\_FORECAST\_PHY\_001\_024 (section 2.1.1), MEDSEA\_ANALYSIS.... (section 2.1.2), IBI\_ANALYSIS... (section 2.1.2). However, the dataset mentioned above provides values just from 2016. Therefore, I supposed the authors used the dataset: GLOBAL\_REANALYSIS\_PHY\_001\_030, MEDSEA\_REANALYSIS\_..., IBI\_REANALYSIS.... In the last case, please check the sentence "(based on a 3DVAR scheme)" in Section 2.1.1. For MED-SEA\_REANALYSIS\_..., it should be substitute with "(based on a OceanVAR scheme)".

8) In the Table 1, please check : i) Column 'MFS', Line 'Resolution':  $1/16^{\circ}$  (~7 km) or  $1/16^{\circ}$  (~5-6 km); ii) Column 'GLO', Line 'Topography': should be "GEBCO8>200 m, ETOPO1<300 m" instead of "GEBCO8<200 m, ETOPO1>300 m"; iii) Column 'IBI', Line 'Topography': "GEBCO, ETOPO, ???" "???". Please replace the ??? with actual values.

9) In the text the variable  $\zeta$ /f is sometimes indicated as the normalized relative vorticity (pg. 8, l. 21) and sometimes as normalized relative vorticity anomaly (pg. 15, l. 11). In the latter case it should be labelled as  $\zeta$ '/f.

10) Pg 3, I. 11: Please add more recent references (Escudier et al., 2016; Pessini et al, 2018) because they deal with eddies properties oin the Algerian basin detected by eddy detection and tracking algorithms.

11) Pg. 3, I. 26 Please check the sentence: "(Results from other sub-regions in the WMED are included in Supp. 2.5)" because in the Supplementary materials I did not find the Supp. 2.5 and in general the "Results from other sub-regions".
12) Pg 13, I. 14-15: Please provide a reference for the sentence: "This sea is the most energetic region of the western Mediterranean".

13) Pg 13, I. 27: I did not understand what the authors mean with "In the WAG the eddy positions are in the deepest waters in the center of the gyre".

14) Pg 3, I. 5 and I. 13: Please substitute )) with )

15) Pg 6, I 4: The subject is missing. The MFS is produced by the Mediterranean Forecasting System (Italy).

16) Pg 8, I 10: Some variables, for example nonlinearity and eddy intensity, are mentioned before being declared. Therefore, I suggest to substitute the sentences from line 9 to 14, at pg. 8, with "these eddy properties are nonlinearity (N=u/c, where c is ....) and eddy intensity (EI=A/L). N provides a measure of ... EI is a potential proxy for the presence of elevated vertical motions (e.g., Frenger et al., 2015; Mason et al., 2017)".

17) Pg 10, I 5: Please substitute Sec. 2.6 with Sec. 3.2.6

18) Pg 20, from line 5 to line 9: Please delete the sentence "Both anomalies T' and S' in the EAG are slightly weaker than those of the WAG.... 50 and 150 m". This is the repetition of the sentence at pg. 20, line 1 to line 4.

19) The text appears to be incomplete. I found many question marks through the text. Please replace them with actual text. Pg 9, I. 9: Supp. ?? Pg 13, I. 13: Supp?? Pg 14, fig. 5: non.inearity in i through p are shown in Supp. ??. Pg 16, I. 26: See Sec. 2.5 and Supp. ?? Pg 22, I. 2: see Figs. ?? and S?? in Supp. ??. Pg 24, I. 24: Supp ?? Pg 24, I. 25: Figs ?? ??

Reference:

Escudier, R., Renault, L., Pascual, A., Brasseur, P., Chelton, D., and Beuvier, J.: Eddy properties in the Western Mediterranean Sea from satellite altimetry and a numerical
simulation, J. Geophys. Res-Oceans, 3990–4006, doi:10.1002/2015JC011371, 2016.

Pessini, F., A. Olita, Y. Cotroneo, and A. Perilli: "Mesoscale eddies in the Algerian Basin: do they differ as a function of their formation site?", Ocean Science, 14, 669–688, https://doi.org/10.5194/os-14-669-2018.

OSD

---

## Author Comment (AC1) · 1 May 2019

We thank the reviewer for the helpful comments and suggestions, that will lead a much improved final manuscript. We also wish to apologise to the reviewer regarding the incomplete state of our initial submission. This we discovered was because we submitted the wrong pdf, and neither we nor OSD picked up on that mistake. The fault is entirely ours. We appreciate that this made reviewing the submission rather challenging for both the reviewers.

In the response below we highlight our replies with ————

[Figure]

Anonymous Referee #1 General Comments: In this study, the authors present an eddy tracking tool to investigate the eddy proper- ties derived from three different MFC re- analysis products. A gridded altimetry dataset is used as reference. The main ar- gument for the added value of this study, is that such a tool is useful for model in- tercomparison, with the objective to evaluate model performance. The manuscript shows some interesting results regarding the eddy properties for the three different MFC reanalysis products, focusing on the western Mediterranean be- tween Gibral- tar and Sardinia, an area governed by intense eddy activity. In my view, this model intercomparison is valid with such a tool and quite interesting. However, without refer- ences to make a proper comparison it is impossible to evaluate model per- formance between the MFCs, let alone to propose remedies for their implementation. This is a major constraint in the way the motivation of this study is posed. I would advise the authors to reshape their motivation, but not necessarily limit their analysis only in the model intercomparison. For instance, I would like to see a more comprehensive analysis for the differences between products, based on attribution to physics and on the knowledge of the region's dynamics, in order to bypass the lack of a reference dataset. They only thing that the authors should be careful is to clearly stress that they make fair assumptions on how they foresee the future improvements for those MFCs, based on the findings of the eddy tracker. After all, some of the co-authors are lead- ing the developments in those MFCs and have a good knowledge of the systems they operate. In the same line of thinking, the draft is too descriptive focusing on the eddy properties and the model intercomparison, without the authors attempting to discuss in details differences in physics between the MFCs products. ————We accept these comments and will work on incorporating these ideas into the revised manuscript.

Finally, one additional problem with the manuscript is that it is sloppily written, as if it was copied/pasted in a rush manner from another draft or report. The figures are not properly introduced in the text and I had to guess throughout the whole review in which figure the authors are referring to. ————This was related to our error with the incorrect pdf, for which we again apologise.

Overall, I find the manuscript worthy of publication in Ocean Science, after a major revision. Please find below a list of comments following the flow of the text, that the authors should address in the revised manuscript.

Specific comments: 1) In this study, an eddy-tracker modelling tool is used to evaluate other models. How confident the authors are that their approach is valid and what are the limits of the method? In the Introduction section I felt that there are not enough references to link this approach with previous studies. ————We are not aware of any close similarities between our work and other previous studies. There are of course many papers that feature eddy compositing (e.g., among others, Chelton etal 2011; Gaube etal 2015; Chaigneau etal 2011), but few of these extend to 3D composites, and none that we know of are applied to model outputs for the purposes of evaluating differences between those models. Perhaps the closest is a paper by Neu et al. (2013) which is an intercomparison of algorithms to detect and track extratropical cyclones; but this article compares the detection/tracking ability rather than the model itself. ————Concerning the validity of our approach, we can say that the tools we have used (i.e., the eddy tracker, and compositing procedure) are robust. But the real test of validity will be whether the respective MFCs take note of our results and find them useful for the ongoing development of their products.

2) In the Abstract, the authors stress the fact that "This information can be informative for the ongoing development". I agree and I was eager to see such an analysis in the Discussion and Conclusion section, but in my view the authors failed to do so. I think that such an analysis is important and the authors should provide a more comprehen- sive and detailed discussion. ————We will address this area in the new manuscript.

3) Page 2, line 16: What is "STA (2017)"? ————This is a citation to a CMEMS technical report; it now appears correctly in the reference list. However we will probably completely remove it and replace it with a reference to a new paper by P.-Y. Le Traon that is in press at Frontiers in Marine Science.

4) Page 3, line 7 and elsewhere in the text for all the figures: "Fig. 1.1". This is not the proper format to mention the figures and should be changed everywhere in the text. In addition, I 've noticed that some figures should have been mentioned earlier in the text with respect to other figures, or other figures are not mentioned at all (e.g. Fig. 2 seems not to be mentioned in the text, unless it is Fig. 2.1, but then should be mentioned earlier). In addition, all figures in the supplement material are referred with question marks, and it was impossible to understand which is which to the point that it become annoying. ————————We apologise for this, and understand the reviewer's frustration. We have been careful to ensure all of these issues are now fixed.

5) Page 3, line 21: "that extend a horizontal distance of 8x the eddy radius". Why? It seems to me that an explanation or a reference is missing here. ————————Changed to "... extend a horizontal distance well beyond the eddy radius". The lines here are intended as a general overview of the compositing procedure; there is a more detailed description later in Sec. 2.4.

6) Page 5, lines 12-13: "Delivered products are bilinearly interpolated onto a regular lon/lat 1/36 grid". This is a major issue for the analysis that follows. Why to interpolate coarser MFS and GLO products into the higher resolution IBI and not leave it as is or make a coarsening of the IBI and MFS to GLO resolution? I guess this is also linked to the tuning of the eddy tracker, but an explanation is missing here and this fact compromises the findings of this work. The authors should elaborate us regarding this pre-processing interpolation. Perhaps, as a supplement material the authors can provide one example for the opposite technique to coarsen IBI and MFS into 1/12 GLO resolution and perform the eddy tracker (with same/different tuning?). ————————This line is incorrect, for which we apologise, and has been removed. There is a horizontal interpolation step in making the eddy composites; this is now better explained in section 2.4 (Eddy compositing).

7) Page 5, lines 13-17: "The stated general...ocean dynamics". This is a very general statement that is redundant in this section. Remove or include it in the Introduction.

—————These lines have been edited and moved to the first paragraph of the sub-section (2.1, The CMEMS models).

8) Table 1 IBI ETOPO definition is presented with question marks and is missing. —————This will be resolved in the new manuscript.

9) Page 8, lines 1-3: "The same... will differ". How valid is the fact that the authors use the same tuning for different resolution products (this is also related to the pre-processing interpolation at 1/36; see my previous comment 6). —————We think this is the correct approach, and have added two further sentences for explanation: "The main implication is that smaller eddies in the higher resolution models, IBI and to a lesser degree, MFS, may not be identified. As our main focus is the mesoscale eddies of the Alboran gyres we do not see this as a significant drawback to our experimental setup." We note also that using different parameters for each model would increase the degrees of freedom, such that it would be harder to attribute differences in the composites to the models or the different choices of tuning parameters.

10) Page 8 in section 2.3: the definitions Le, A and EI should all be declared one line earlier I think. —————Done, and reads much better now.

11) Page 8 section 2.4: Is the composite grid at 1/36? Calculations of dynamical properties (e.g. vorticity) are first calculated on the native MFC regular grids as delivered through the CMEMS portal and then on 1/36? Clarify in the text. Units for the composite grid can be considered as fractions of the eddy radii. This is now explained clearly in Sec.2.4, with reference to Figure 2. —————Vorticity is calculated on the native regular grid of each model, and then interpolated to the compositing grid just as for the other variables (temperature, salinity, etc.)

12) Page 10, line 5: "...and their range variability is shown in Sec. 2.6". We are in Sec. 2.6, what I have missed here? —————This link has been corrected, it now points to sec 3.2.6 "Seasonal mixed layer depth".

13) Page 13, line 25: "...property anomalies of the...". Anomalies with respect to what? Clarify in the text. The term "anomalies" is also used elsewhere in the text and a clarification is needed there as well. —————"Anomalies" has been removed.

14) Pages 15-16 "The model increase can be explained by the increasing model resolution". Bad phrasing. Rephrase in the text. —————This has been rewritten as "For each model, ${|\zeta|}$ intensity in each subregion successively decreases from the west (WAG) to east (CRT). There is also an overall increase in ${|\zeta|}$ in each subregion between the models; \emph{GLO} is weakest and \emph{IBI} strongest.".

15) Page 16, lines 2-3: "in IBI is suspected...may be an overestimate". Why do you suspect that, is there any reference or an explanation? —————We have added a reference to our CMEMS MedSub project report that shows a very high EKE in IBI. Further, our coauthor Marcos Garcia has stated that the jet is suspected to be too strong in the IBI version we use in this paper.

16) Page 16, line 26: "...at each vertical level". Is there some sort of vertical re-gridding in the post-processing for the different products pertaining to the fact that the MFCs have different vertical grids? Clarify in the text. —————No, regridding is not necessary. We have modified the line as follows: "tilt is estimated based on the position of absolute maximum $\zeta$ within the eddy radius at each vertical $z$-level for the respective models." and in section 2.5 we added the statement "No interpolation or vertical re-gridding is required.".

17) Page 18, line 2 (and Table 2): "...are around 9% smaller than the means...". The radii median is always found smaller than the radii mean. Does this means something for the eddies properties, shape and distribution? —————We will address this in the revised manuscript.

18) Page 22, lines 19-20: "As shortwave solar radiation increases from spring to summer (Ruiz et al., 2008),...". Do we really need a reference for this statement? ————–We removed this reference.
19) Page 24, lines 15-17: "The most likely hypothesis...acts to cool the surface mixed layer". Why not Data Assimilation, how can you tell? How do we know that IBI has the best performance in terms of upper ocean temperature due to tides, when the other MFCs include SST DA? I guess you should rephrase this part of the text and include DA in your fair assumptions. In addition, in the continuation of this paragraph the authors attempt to make some recommendations to MFCs, based on tides and bulk heat fluxes. I am OK with that, but I think there are also other factors that should be mentioned here, such as vertical parametrizations, turbulent closer schemes, vertical stretching and number of levels (especially when the MFS has increased by a lot the number of vertical levels in the new version, hinting perhaps to better performance?) etc. Finally, suggesting that the GLO MFC should have tides is a logical but not straightforward suggestion, because of the complexity in doing so, and because the area under investigation in this study is very small to derive any strong conclusion for the global configuration. —————These are all valid comments and we are working with our coauthors to address them in the new version.

20) Regarding the authors comment for the ARMOR3D product in the Discussion and Conclusion section, I agree that without such products is difficult to assess how close to reality the eddy composites are. However, even if you had this product on its coarse resolution (if not mistaken at 1/4 of a degree?), that would still not be sufficient for your domain of application, so I guess the ARMOR3D product for regional applications should be delivered on a higher resolution grid? Perhaps the authors want to comment on this. —————Yes, it is $\frac{1}{4}$ degree and you are correct that it would be too small to be of use in the Alboran Sea. We have added some lines in the new text to address this issue in more detail.

---

## Author Comment (AC2) · 1 May 2019

We thank the reviewer for the their helpful input. We also wish to apologise to the reviewer regarding the incomplete state of our initial submission. This we discovered was because we submitted the wrong pdf, and neither we nor OSD picked up on that mistake. The fault is entirely ours. We appreciate that this made reviewing the submission rather challenging for both the reviewers.

We mark our replies below with ————

Anonymous Referee #2 General Comments: In

[Figure]

this manuscript, the authors applied an eddy tracker and an eddy compositing technique, to three forecasts models: (i) the Mediterranean Forecasting System (MFS), (ii) the Global Mercator model (GLO ), and (iii) the Iberia-Biscay-Ireland system (IBI ), provided by the Copernicus Marine Service (CMEMS) website. They used the py-eddy tracker, an eddy detection and tracking method similar to the procedure proposed by Chelton et al. (2011). This method is widely used by the scientific community. The eddy compositing analysis is applied to analyse the 3-D structures of three gyres of the Alboran Sea (WAG, EAG, CRT), an interesting area of the Western Mediterranean, characterized by strong density gradients and high mesoscale activity. The authors compared the models results to evaluate their performance and conclude the paper with some technical suggestions in order to improve these CMEMS products. This paper is well structured and the methodological approach is correct for the purpose of the journal. Nevertheless a more detailed description of the methods and additional results are needed to support the interpretation and conclusion. Therefore, at this stage, a major revision is required before acceptance. Please see below for more details:

Major remarks: 1) I found the Section 2.4 (Eddy compositing) hard to follow. The methodology used for the 3-D eddy composites and for the computation of the anomalies of T and S is not adequately explained. The authors refer to the article of Mason et al., 2017 (see pg. 8 line 15) but it is applied in a different context and in a different area. ————A full description is now included, which includes reference to figure 2.

2) The eddy properties, detected by py-eddy tracker applied to the three models and to the ALT data, are not adequately compared (section 3.1). In Section 3.1.1, the results on the numbers of eddies detected and tracked in relation to their lifetimes are not quantified. In Section 3.1.2, the authors have identified similarities "between the patterns" of eddy radius, amplitude and intensity (pg. 10, l.16, pg. 11, l.1 and l.11). These may be due to the strong signal of the Atlantic Water flowing eastward, which generates the large and energetic Anticyclonic Eddies. This flow causes differences in terms of eddy properties between the northern and the southern part of the basin.

Pessini et. al, 2018 identified these N-S differences as function of the area of formation and lifetime. Escudier et al., 2016 compared the eddies distribution per radius and lifetime, of altimetry and model data. I suggest the authors to follow the last approach since, in my opinion, more detailed comparison between the models and ALT data are needed. —————We have added several lines to the Introduction about the reasons for the N-S differences in the WMED.

3) The authors did not describe how they calculated the mean and median coordinates/radii of the eddies, listed in Table 2. There are various methods to calculate them for example: (i) selecting all the eddies detected by the eddy tracker independently for lifetime, (ii) discriminating for lifetime (i.e. avoiding the shortest) and (iii) selecting only the eddies identified simultaneously by the three models in the same sub-area. The 3-D eddy composite analysis should depend on these variables (pg.3, l. 10 to l. 22) and therefore should be method-dependent. Some results (pg. 18, l. 11 to l.12) and conclusions (pg. 24, l. 13 to l. 21) are based on vertical structure of T'. The authors ac- tually cannot demonstrate "how closely these eddy composite results may correspond to reality" (pg. 26, l.10-11). For these reasons, a check to verify if the 3-D subregion eddy composite results are method-dependent should be done. Alternatively, I suggest the third method because is the most appropriate to compare the model results. —————These statistics were calculated using the second method suggested by the reviewer, where the minimum eddy lifetime (5 days) is set as an input parameter to the eddy tracker. The third method is not practical because IBI does not have data assimilation so there is no meaningful correspondence at all with altimetric eddies. Lastly, we are working on providing a reference for the composite T and S profiles using Argo floats.

4) In Table 2, please provide the mean and median coordinates and the properties of the eddy for the altimetry data (section 3.2). This will allow a comparison among the models and the altimetry data. —————The altimetry information has been added to the table.

5) Table 3 is not acknowledge through the text. The variables Rmin, Rmax, Rmean, Rmedian, Rmad are not defined. Some values are missing and other seem to have no sense. Specifically, in "Product: MFS" at line "CRT" are listed the values: 111, 222,333, 444, 555 which seem to be out of scale. In Section 2 (Data and methods) the effective and the speed-based (inner) radii are indicated respectively with the symbols Le and L while in Table 3 are labelled as R. ————Table 3 has been removed as it was redundant.

6) In Figure 10, the seasons are not mentioned. In winter and autumn, the deepening in the mixed layer depth (MLD) in the center of the anticyclonic eddy increase from GLO to IBI (fig. 10). This seems coherent with the zonal and meridional relative vorticity of figures 7 and 8, having more intense anomaly for higher resolution models. I assumed that these differences in the MLD inside the anticyclonic eddy are due to rotational speeds, better simulated in these higher resolution models. Anyway the authors assert that "the incoming Atlantic Jet in IBI is suspected to be too strong such that these $\zeta$ values may be an overestimate (pg. 16, l. 2-3)". The last sentence should be motivated and this part must be clarified also because these considerations could be used to evaluate the performance of the models (pg. 24, l. 22) and to provide a more consistent conclusion. ————The text in the MLD section (3.2.6) has been modified. Labels for the seasons in figure 10 have been added. Concerning the statements about the Atlantic Jet in IBI, we added a reference to the CMEMS MedSub project report which contains more information about this anomaly.

Minor remarks: 7) In Section 2.1 (pg. 5, l. 10-11), the authors state that the variables, for the period 1 January 2013 to 30 June 2016, are downloaded by ftp from the Coper- nicus CMEMS portal from: GLOBAL_ANALYSIS_FORECAST_PHY_001_024 (sec- tion 2.1.1), MEDSEA_ANALYSIS... . . (section 2.1.2), IBI_ANALYSIS. . . . (section 2.1.2). However, the dataset mentioned above pro- vides values just from 2016. Therefore, I supposed the authors used the dataset: GLOBAL_REANALYSIS_PHY_001_030, MEDSEA_REANALYSIS_... ,

[Figure]

IBI_REANALYSIS. . . . In the last case, please check the sentence "(based on a 3DVAR scheme)" in Section 2.1.1. For MED- SEA_REANALYSIS_..., it should be substitute with "(based on a OceanVAR scheme)". ————No, we did not use the suggested *REANALYSIS* products. The products we used were the ones that were available at the start of the MedSub project in March 2016. The GLOBAL* _001_024 data are still available for the period 2006-2016 by using the motu_client software, rather than using the subsetting option on the CMEMS portal. ————We changed the text to "based on an OceanVAR scheme" as suggested.

8) In the Table 1, please check : i) Column 'MFS' , Line 'Resolution': 1/16âŮę (âĹij7 km) or 1/16âŮę (âĹij5-6 km); ii) Column 'GLO', Line 'Topography': should be "GEBCO8>200 m, ETOPO1<300 m" instead of "GEBCO8<200 m, ETOPO1>300 m"; iii) Column 'IBI', Line 'Topography': "GEBCO, ETOPO, ???" "???". Please replace the ??? with actual Values. ————We believe what we have is correct regarding the topography; the information was taken from the CMEMS GLO website. We will confirm this in the new manuscript.

9) In the text the variable $\zeta/f$ is sometimes indicated as the normalized relative vorticity (pg. 8, l. 21) and sometimes as normalized relative vorticity anomaly (pg. 15, l. 11). In the latter case it should be labelled as $\zeta'/f$. ————'Anomaly' has been removed in all cases.

10) Pg 3, l. 11: Please add more recent references (Escudier et al., 2016; Pessini et al, 2018) because they deal with eddies properties in the Algerian basin detected by eddy detection and tracking algorithms. ————These references have been added.

11) Pg. 3, l. 26 Please check the sentence: "(Results from other sub-regions in the WMED are included in Supp. 2.5)" because in the Supplementary materials I did not find the Supp. 2.5 and in general the "Results from other sub-regions". ————These will appear in the new manuscript.

12) Pg 13, l. 14-15: Please provide a reference for the sentence: "This sea is the

most energetic region of the western Mediterranean". ————We added a reference to Pascual etal 2007 and Capó etal JPO 2019.

13) Pg 13, l. 27: I did not understand what the authors mean with "In the WAG the eddy positions are in the deepest waters in the center of the gyre". ————We changed this sentence to: "In the WAG, the median eddy positions for each model are located in the center of the gyre, which corresponds to the deepest water."

14) Pg 3, l. 5 and l. 13: Please substitute )) with ) ————Done.

15) Pg 6, l 4: The subject is missing. The MFS is produced by the Mediterranean Forecasting System (Italy). ————This has been corrected.

16) Pg 8, l 10: Some variables, for example nonlinearity and eddy intensity, are mentioned before being declared. Therefore, I suggest to substitute the sentences from line 9 to 14, at pg. 8, with "these eddy properties are nonlinearity (N=u/c , where c is . . ..) and eddy intensity (EI=A/L). N provides a measure of . . . EI is a potential proxy for the presence of elevated vertical motions (e.g., Frenger et al., 2015; Mason et al., 2017)". ————Done as suggested.

17) Pg 10, l 5: Please substitute Sec. 2.6 with Sec. 3.2.6 ————Done.

18) Pg 20, from line 5 to line 9: Please delete the sentence "Both anomalies T' and S' in the EAG are slightly weaker than those of the WAG. . .. . . .. 50 and 150 m". This is the repetition of the sentence at pg. 20, line 1 to line 4. ————Corrected as suggested.

19) The text appears to be incomplete. I found many question marks through the text. Please replace them with actual text. Pg 9, l. 9: Supp. ?? Pg 13, l. 13: Supp?? Pg 14, fig. 5: nonlinearity in i through p are shown in Supp. ??. Pg 16, l. 26: See Sec. 2.5 and Supp. ?? Pg 22, l. 2: see Figs. ?? and S?? in Supp. ??. Pg 24, l. 24: Supp ?? Pg 24, l. 25: Figs ?? ?? ————The text was indeed incomplete owing to our error with the pdf. These issue will all be corrected in the new manuscript.

References

Capó, E., A. Orfila, E. Mason, and S. Ruiz, 2019: Energy Conversion Routes in the Western Mediterranean Sea Estimated from Eddy–Mean Flow Interactions. J. Phys. Oceanogr., 49, 247–267, https://doi.org/10.1175/JPO-D-18-0036.1

Pascual, A., Marie-Isabelle Pujol, Gilles Larnicol, Pierre-Yves Le Traon, Marie-Hélène Rio, Mesoscale mapping capabilities of multisatellite altimeter missions: First results with real data in the Mediterranean Sea, Journal of Marine Systems, Volume 65, Issues 1–4, 2007, Pages 190-211, https://doi.org/10.1016/j.jmarsys.2004.12.004.